# ZeroPatcher: Training-free Sampler for Video Inpainting and Editing

Shaoshu Yang[1,3], Yingya Zhang[2], and Ran He[1,3]

[1]School of Artificial Intelligence, University of Chinese Academy of Sciences
[2]Tongyi Lab
[3]New Laboratory of Pattern Recognition (NLPR), CASIA

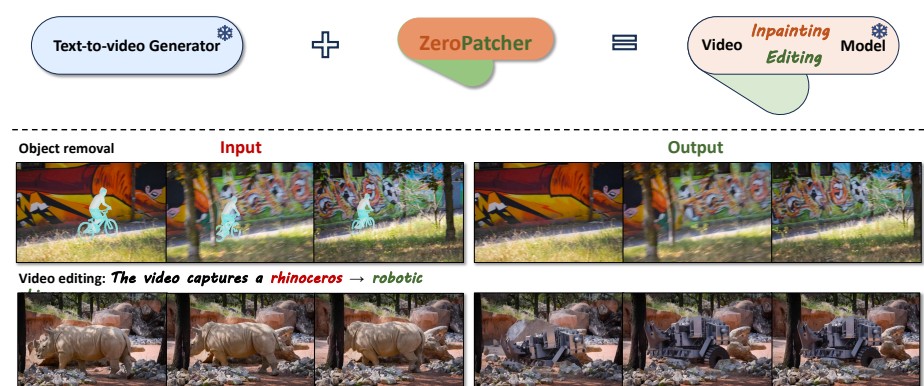

Figure 1: With ZeroPatcher, a text-to-video generative model can achieve video inpainting and video editing without any finetuning.

## Abstract

Video inpainting and editing have long been challenging tasks in the video generation community, requiring extensive computational resources and large datasets to train models with satisfactory performance. Recent breakthroughs in large-scale video foundation models have greatly enhanced text-to-video generation capabilities. This naturally leads to the idea of leveraging the prior knowledge from these powerful generators to facilitate video inpainting and editing. In this work, we investigate the feasibility of employing pre-trained text-to-video foundation models for high-quality video inpainting and editing without additional training. Specifically, we introduce a model-agnostic denoising sampler that optimizes the trajectory by maximizing the log-likelihood expectation conditioned on the known video segments. To enable efficient dynamic object removal and replacement, we propose a latent mask fuser that performs accurate video masking directly in latent space, eliminating the need for explicit VAE decoding and encoding. We implement our approach in widely-used foundation generators such as CogVideoX and HunyuanVideo, demonstrating the model-agnostic nature of our sampler. Comprehensive quantitative and qualitative evaluations confirm that our method achieves outstanding video inpainting and editing performance in a plug-and-play fashion.

## 1 Introduction

Given an input video and a dynamic mask, video inpainting or editing tasks require models to render the masked regions according to user specifications. As a long-standing challenge in video research,

39th Conference on Neural Information Processing Systems (NeurIPS 2025).

this problem has been approached through various paradigms, including transformers (44; 17) and diffusion models (42; 15). Recent advances in foundational video generators (33; 38; 28), particularly through large diffusion transformers, have significantly improved video generation capabilities. This progress naturally suggests leveraging these powerful foundation models to advance video inpainting and editing. However, effectively utilizing their conditional generation abilities for these tasks would typically demand substantial computational resources for training, given their massive scale. Furthermore, as foundation models continue to evolve, traditional approaches relying on extensive fine-tuning will face increasing challenges in adapting to new video generators. An alternative solution is to employ these video generators as data priors, enabling task resolution in a training-free manner.

Recent research has extensively explored methods for enabling conditional generation in image diffusion models. For inpainting tasks where significant portions of the image are unavailable, current approaches typically employ back-projection operations (32) to incorporate information from known regions. However, this process does not directly estimate the conditional denoising distribution and may fail when the generation trajectory substantially deviates from the known data (32).

In this work, we introduce ZeroPatcher, a novel training-free approach that unlocks video inpainting capabilities in text-to-video foundation models. Our method is theoretically model-agnostic and compatible with diffusion-based video generators. To approximate the conditional denoising distribution, we propose *conditional denoising expectation maximization* (CD-EM), which formulates an EM problem over the denoising trajectory conditioned on known data. The framework consists of an expectation step implemented through Monte-Carlo sampling and a maximization step solved via fixed-point iteration. We further establish the uniqueness of the fixed point in the maximization step through theoretical analysis. While existing methods inject known information by pixel replacement at each denoising step, this approach is incompatible with prevalent latent diffusion models that cannot perform precise masking in latent space. To overcome this limitation, we present a *mask fuser* - a lightweight convolutional architecture that enables accurate dynamic masking in latent space.

We perform extensive evaluations on the DAVIS (22) and YouTube-VOS (35) datasets to assess both the inpainting and editing capabilities of our method. By leveraging the generative power of modern video foundation models, our approach achieves performance competitive with trained inpainting models. Furthermore, we demonstrate ZeroPatcher's video editing potential, showing superior ability to modify object shapes while maintaining surrounding content compared to existing methods. Our key contributions are:

- We introduce ZeroPatcher, a training-free framework that adapts video diffusion models for video inpainting and editing. The proposed latent mask fuser enables dynamic video masking directly in latent space.

- We present *conditional denoising expectation maximization* (CD-EM) to optimize sampling trajectories using known video regions as guidance, accompanied by theoretical analysis proving solution uniqueness and convergence.

- Comprehensive experiments demonstrate our method's model-agnostic nature and competitive performance against trained approaches across various video inpainting and editing tasks.

## 2 Related Works

### 2.1 Video generative models

Recent advances in generative learning methods, including autoregressive models (29), diffusion models (32; 26; 21), and flow matching models (19), have significantly enhanced video generation capabilities. Early approaches focused on learning video distributions in pixel space, with Video Diffusion Model (12) pioneering the application of diffusion denoising to this domain. Subsequent works like Make-A-Video (25), PYoCo (8), and Imagen Video (10) integrated large language models to enable text-to-video generation. To address the high dimensionality of video data, recent research has shifted toward latent space learning. VideoGPT (37), combining VQ-VAE (6) with transformer-based next-token prediction, established an early benchmark for latent video modeling. He et al. further advanced this direction by employing diffusion models to approximate latent video distributions, giving rise to latent video diffusion models (9; 43; 34; 2; 1; 31; 3; 4). CogVideoX (38)

introduced diffusion transformers and temporal-compressed video VAEs, substantially improving motion complexity. With growing datasets and computational resources, large-scale video generators (28; 33) are now achieving unprecedented video generation quality.

## 2.2 Video inpainting and editing

Video inpainting and editing represent crucial downstream applications in video generation research. For video inpainting, E2FGVI (16) employs optical flow to guide the inpainting process, while ProPainter (44) utilizes a transformer architecture to expand the perceptual field. More recently, Zhang et al. (42) introduced diffusion models to video inpainting by training a UNet for masked conditional denoising. The rapid advancement of foundation models has similarly propelled progress in video editing. Approaches like FateZero (23) and VideoP2P (18) leverage diffusion inversion and attention map control for content swapping while preserving backgrounds, whereas VideoComposer (30) trains an editing-compatible generator. However, these methods rely on architecture-specific properties. In this work, we investigate a model-agnostic approach that enables text-to-video generators to perform inpainting and editing through theoretical sampling analysis.

# 3 Method

## 3.1 Preliminatry

**Diffusion model**   Given a video $\mathbf{x}_0 \sim p(\mathbf{x}_0), \mathbf{x}_0 \in \mathbb{R}^{t \times h \times w \times c}$, where $t, h, w$ are temporal and spatial dimensions and $c$ is the latent channel size. Generative models approximate this distribution through a series of Markovian diffusion and denoising transitions (12; 26; 21; 27). The forward and backward diffusion processes are defined as:

$$q(\mathbf{x}_t|\mathbf{x}_0) = \mathcal{N}(\mathbf{x}_t|\sqrt{\alpha_t}\mathbf{x}_0, (1-\alpha)\mathbf{I}) \qquad (1)$$

$$q(\mathbf{x}_{t-1}|\mathbf{x}_t, \mathbf{x}_0) = \mathcal{N}(\mathbf{x}_{t-1}|\boldsymbol{\mu}(\mathbf{x}_t, \mathbf{x}_0), \sigma_t^2\mathbf{I}), \qquad (2)$$

where $\mathbf{x}_t$ denotes the noisy data at timestep $t = 1, 2, \cdots, T$, and the mean $\boldsymbol{\mu}(\mathbf{x}_t, \mathbf{x}_0)$ is a linear combination of $\mathbf{x}_t$ and $\mathbf{x}_0$. A diffusion model is trained to iteratively denoise samples starting from Gaussian noise $\mathbf{x}_T \sim \mathcal{N}(\mathbf{x}_T|\mathbf{0}, \mathbf{I})$. The model learns to approximate the reverse diffusion process conditioned on $y$ (e.g. textual prompts) through the distribution $p_\theta(\mathbf{x}_{t-1}|\mathbf{x}_t, y) = \mathcal{N}(\mathbf{x}_{t-1}|\boldsymbol{\mu}_\theta(\mathbf{x}_t, y), \boldsymbol{\Sigma}_\theta(\mathbf{x}_t), y)$. For notational simplicity, we will generally omit the explicit conditioning on $y$ in subsequent sections. The diffusion model is parameterized to predict the clean data $\mathbf{f}_\theta(\mathbf{x}_t) \simeq \mathbf{x}_0$. Using this trained model, we can approximate the denoising distribution by substituting the ground truth $\mathbf{x}_0$ with the model's prediction: $p_\theta(\mathbf{x}_{t-1}|\mathbf{x}_t) = q(\mathbf{x}_{t-1}|\mathbf{x}_t, \mathbf{f}_\theta(\mathbf{x}_t))$. Specifically, the learned denoising distribution is

$$p_\theta(\mathbf{x}_{t-1}|\mathbf{x}_t) = \mathcal{N}(\mathbf{x}_{t-1}|\boldsymbol{\mu}(\mathbf{x}_t, \mathbf{f}_\theta(\mathbf{x}_t)), \sigma_t^2\mathbf{I}). \qquad (3)$$

**Training-free image inpainting**   The back-projection method (32; 7) has emerged as a prevalent approach for zero-shot image inpainting. This method employs a linear degradation operator $\mathbf{A}$, where the degraded observation is given by $\mathbf{x}_0^a = \mathbf{A}^\dagger \mathbf{A} \mathbf{x}_0$. Common degradation operators include masking, downsampling, and monochromatization. The complete image can be decomposed into orthogonal components:

$$\mathbf{x}_0 = \underbrace{\mathbf{A}^\dagger \mathbf{A} \mathbf{x}_0}_{\text{rangespace}} + \underbrace{(\mathbf{I} - \mathbf{A}^\dagger \mathbf{A})\mathbf{x}_0}_{\text{nullspace}}, \qquad (4)$$

where $\mathbf{A}^\dagger$ is the pseudo inverse of $\mathbf{A}$ such that $\mathbf{A}\mathbf{A}^\dagger\mathbf{A} = \mathbf{A}$. The range space component represents the preserved information in the degraded observation, while the null space component contains the missing information to be recovered. In training-free inpainting, we assume access to $\mathbf{x}_0^a$ but not the original $\mathbf{x}_0$. The key idea is to leverage a pretrained diffusion model $\mathbf{f}_\theta(\cdot)$ to estimate the null space component during denoising. DDNM (32) proposes a back-projection step that enforces consistency with the observed data:

$$\hat{\mathbf{f}}_\theta(\mathbf{x_t}) = (\mathbf{I} - \mathbf{A}^\dagger \mathbf{A})\mathbf{f}_\theta(\mathbf{x}_t) + \mathbf{A}^\dagger \mathbf{x}_0^a. \qquad (5)$$

This formulation ensures the diffusion model only predicts in the null space of $\mathbf{A}$ while perfectly preserving the range space component $\mathbf{x}_0^a$ in the final output.

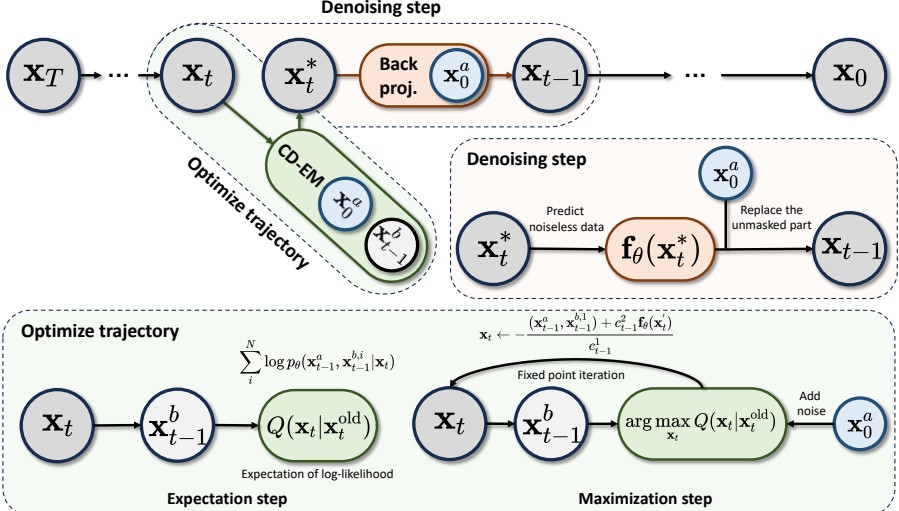

Figure 2: An illustration of the sampling process with *conditional denoising expectation maximization* (CD-EM) and back-projection method. In each sampling step, we first use CD-EM to optimize the sampling trajectory with the guidance of the known data. Then we inject known information through back-projection method.

## 3.2    Formulation

Consider a data instance $\mathbf{x}_0 = \{\mathbf{x}_0^a, \mathbf{x}_0^b\}, \mathbf{x}_0 \in \mathbb{R}^d$ composed of masked $\mathbf{x}_0^b$ and unmasked components $\mathbf{x}_0^a$, where the partitioning is defined by a binary mask $\mathbf{m} \in \mathbb{Z}^d, \mathbf{m}_i \in \{0, 1\}, \mathbf{x}_0^a = \{\mathbf{x}_0^i | \mathbf{m}_i = 0\}, \mathbf{x}_0^b = \{\mathbf{x}_0^i | \mathbf{m}_i = 1\}$. This mask-based partitioning extends naturally to noisy data $\mathbf{x}_t$ at any timestep $t$. Given a pretrained diffusion model $\mathbf{f}_\theta(\cdot)$ that estimates clean data from noisy inputs $\mathbf{f}_\theta(\mathbf{x}_t)$, our goal is to recover $\mathbf{x}_0^b$ through conditional denoising while preserving $\mathbf{x}_0^a$ starting from $\mathbf{x}_T$.

## 3.3    Conditional denoising expectation maximization

For inpainting tasks, we observe the ground truth denoising trajectory for known segments $\boldsymbol{x}_t^a$. Current back-projection methods sample from $p_\theta(\mathbf{x}_{t-1} | \mathbf{x}_t^a, \mathbf{x}_t^b)$ (32; 12), where $\boldsymbol{x}_t^b$ represents noisy inpainting results. However, this approach suffers from two key limitations:

- The estimation is biased from the true conditional denoising distribution $p_\theta(\mathbf{x}_{t-1} | \mathbf{x}_t, \mathbf{x}_0^a)$. While diffusion models approximate the unconditional distribution $p_\theta(\mathbf{x}_{t-1} | \mathbf{x}_t)$. learning the conditional variant requires additional training.
- The method only leverages information from $\boldsymbol{x}_t^a$ at noise level $t$, ignoring less noisy versions (e.g. $\boldsymbol{x}_{t-1}^a$) that contain more reliable information.

Since pretrained text-to-video diffusion models lack explicit conditional generation capabilities, we circumvent the need for $p_\theta(\mathbf{x}_{t-1} | \mathbf{x}_t, \mathbf{x}_0^a)$ by reformulating the sampling process as a maximum likelihood estimation problem. The pipeline of the method is presented in fig. 2. Our proposed *conditional denoising expectation maximization* (CD-EM) method optimizes the sampling trajectory to maximize the joint likelihood We aim to follow a sampling trajectory that maximizes the likelihood $p_\theta(\mathbf{x}_{t-1}^a, \mathbf{x}_{t-1}^b | \mathbf{x}_t)$. Within this likelihood, $\boldsymbol{x}_{t-1}^b$ denotes the unknown masked component. While $\mathbf{x}_{t-1}^a = \sqrt{\alpha_{t-1}} \mathbf{x}_0^a + \sqrt{1 - \alpha_{t-1}} \epsilon_{t-1}$ represents the known unmasked ones. The expectation of the log-likelihood is defined as:

$$Q(\mathbf{x}_t | \mathbf{x}_t^{\mathrm{old}}) = \int_{\mathbf{x}_{t-1}^b} p_\theta(\mathbf{x}_{t-1}^b | \mathbf{x}_t^{\mathrm{old}}) \log p_\theta(\mathbf{x}_{t-1}^a, \mathbf{x}_{t-1}^b | \mathbf{x}_t) d\mathbf{x}_{t-1}^b \tag{6}$$

We use $\mathbf{x}_t^{\mathrm{old}}$ to denote the old noisy latent in the previous EM step. In eq. (6), sampling from $p_\theta(\mathbf{x}_{t-1}^b | \mathbf{x}_t^{\mathrm{old}})$ requires neural network forward propagation and thus the integration is intractable.

We can approximate it with Monte-Carlo integration:

$$Q(\mathbf{x}_t | \mathbf{x}_t^{\text{old}}) \simeq \sum_i^N \log p_\theta(\mathbf{x}_{t-1}^a, \mathbf{x}_{t-1}^{b,i} | \mathbf{x}_t), \tag{7}$$

where $\mathbf{x}_{t-1}^{b,i}$ is sampled from $p_\theta(\mathbf{x}_{t-1}^b | \mathbf{x}_t^{\text{old}})$ and $N$ is the number of samples. Since the likelihood term is Gaussian. We analytically express the approximated expectation and our objective:

$$Q(\mathbf{x}_t | \mathbf{x}_t^{\text{old}}) \simeq -\frac{1}{2\sigma_{t-1}^2} \sum_i^N \|(\mathbf{x}_{t-1}^a, \mathbf{x}_{t-1}^{b,i}) - \boldsymbol{\mu}(\mathbf{x}_t, \mathbf{f}_\theta(\mathbf{x}_t))\|_2^2 \tag{8}$$

$$\mathbf{x}_t^* = \arg\min_{\mathbf{x}_t} \|(\mathbf{x}_{t-1}^a, \mathbf{x}_{t-1}^{b,i}) - \boldsymbol{\mu}(\mathbf{x}_t, \mathbf{f}_\theta(\mathbf{x}_t))\|_2^2. \tag{9}$$

Given this, we can optimize the conditional denoising trajectory by iteratively computing and maximizing the expectation. This method demonstrates two advantages:

- We does not need to spend additional training to approximate $p_\theta(\mathbf{x}_{t-1} | \mathbf{x}_t, \mathbf{x}_0^a)$. Therefore, our method is training-free and model-agnostic.
- Instead of using $\mathbf{x}_t^a$ as the condition for the denoising step. CD-EM utilizes $\mathbf{x}_{t-1}^a$ which is less noisy and therefore provides a more reliable guidance.

### 3.4 Solving the maximization

In this section, we present our approach to maximize the expectation in eq. (9). The objective is to find a $\mathbf{x}_t$ that satisfies:

$$(\mathbf{x}_{t-1}^a, \mathbf{x}_{t-1}^{b,i}) - \boldsymbol{\mu}_{\boldsymbol{\theta}}(\mathbf{x}_t) = (\mathbf{x}_{t-1}^a, \mathbf{x}_{t-1}^{b,i}) + c_{t-1}^1 \mathbf{x}_t + c_{t-1}^2 \mathbf{f}_{\boldsymbol{\theta}}(\mathbf{x}_t) = 0, \tag{10}$$

where $c_{t-1}^1, c_{t-1}^2$ are constant coefficients. Since this equation cannot be solved analytically due to the neural network function $\mathbf{f}_{\boldsymbol{\theta}}(\cdot)$. In alternative, we propose to employ fixed point iteration to obtain an approximate solution. A special case is when $N = 1$, the iteration formula becomes:

$$\mathbf{x}_t \leftarrow -\frac{(\mathbf{x}_{t-1}^a, \mathbf{x}_{t-1}^{b,1}) + c_{t-1}^2 \mathbf{f}_{\boldsymbol{\theta}}(\mathbf{x}_t')}{c_{t-1}^1}. \tag{11}$$

We find $N = 1$ is able to get plausible results while maintaining computational efficiency. The complete deduction and the matrix formation when $N > 1$ are provided in the appendix. The effectiveness of our fixed point iteration method depends on solution uniqueness and convergence. We establish these properties if the used diffusion model is well trained and $N = 1$.

**Theorem 3.1.** *Given a diffusion model $\mathbf{f}_\theta(\cdot)$, a noisy latent $\mathbf{x}_t$, and a future step known data $\mathbf{x}_{t-1}^a$. If the diffusion model is well trained with Lipschitz constant $\mathcal{L}_s$. The score function $\mathbf{s}(\cdot)$ is approximated through $\mathbf{s}_\theta(\mathbf{x}_t) \simeq \mathbf{s}(\mathbf{x}_t | \mathbf{x}_0)$. The approximation satisfies*

$$\|\mathbf{s}_\theta(\mathbf{x}_t^1) - \mathbf{s}_\theta(\mathbf{x}_t^2)\|_2 < \mathcal{L}_s \|\mathbf{x}_t^1 - \mathbf{x}_t^2\|_2, \tag{12}$$

*where $\mathbf{x}_t^1 \in \mathbb{R}^d$ and $\mathbf{x}_t^2 \in \mathbb{R}^d$ are two arbitrary noisy samples. Then there is a noise schedule $\{\alpha_0, \cdots, \alpha_T\}, \{\sigma_0, \cdots, \sigma_T\}$ and an ending timestep $t_0$. For $T \geq t \geq t_0$, the iteration $\mathbf{x}_t \leftarrow -\left((\mathbf{x}_{t-1}^a, \mathbf{x}_{t-1}^{b,1}) + c_{t-1}^2 \mathbf{f}_{\boldsymbol{\theta}}(\mathbf{x}_t)\right)/c_{t-1}^1$ converges to a unique fixed point.*

The proof of theorem 3.1 is shown in the appendix. This theorem guarantees that CD-EM can effectively optimize the noisy latent $\mathbf{x}_t$ through fixed point iteration. Let $K$ denote the number of fixed point iterations and let $P$ denote the number of EM iterations. While our method involves both Monte Carlo integration and fixed point iteration, experiments demonstrate that high-quality, consistent results can be achieved with $N = 1, K = 1, P = 2$ without significant computational overhead. Notably, when using 1 step of fixed point iteration, no additional neural network forward passes are required.

### 3.5 Sampling with CD-EM

Our method operates independently or with back projection techniques (32). As shown in fig. 2, back projection follows CD-EM optimization. Let $\mathbf{x}_t^*$ denote the CD-EM optimized latent. We then apply

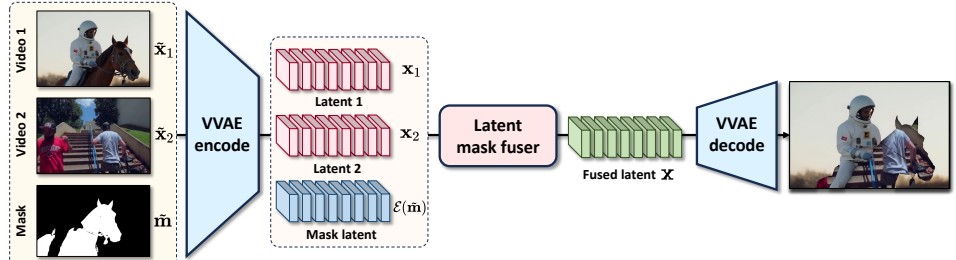

Figure 3: Sketched pipeline of latent mask fuser. We use VVAE to abbreviate video VAE. The model use two videos to be fused and the pixel space dynamic mask as the input. After encoding them with the encoder, the latent mask fuser concatenates the three inputs in channel dimension and perform video fusing in latent space.

denoising $\mathbf{x}_{t-1} \sim p_\theta(\mathbf{x}_{t-1}|\mathbf{x}_t^*)$. For CD-EM only, we optimize $\mathbf{x}_{t-1}$ to $\mathbf{x}_{t-1}^*$. Repeat this process until we reach $\mathbf{x}_0$. With back projection, we inject range space data during sampling. Mathematically, the process is

$$\mathbf{x}_{t-1} \sim \begin{cases} q(\mathbf{x}_{t-1}|\mathbf{x}_t^*, \hat{\mathbf{f}}_\theta(\mathbf{x}_t^*)) & \text{Use back proj.} \\ p_\theta(\mathbf{x}_{t-1}|\mathbf{x}_t^*) & \text{Others} \end{cases} \tag{13}$$

$$\hat{\mathbf{f}}_\theta(\mathbf{x}_t^*) = \{\mathbf{x}_0^a, \mathbf{f}_\theta^b(\mathbf{x}_t^*)\}, \tag{14}$$

where $\mathbf{f}_\theta^b(\mathbf{x}_t^*) = \{\mathbf{f}_\theta^i(\mathbf{x}_t^*)|\mathbf{m}^i = 1\}$ is the masked portion of $\mathbf{f}_\theta(\mathbf{x}_t^*)$. CD-EM shows significant improvements in early denoising stages when visual layouts are determined, with subtle effects in later stages. We suspend CD-EM when $t < \tau$ to save computation (see appendix for algorithm).

### 3.6 Latent mask fuser

Pixel-space masks $\mathbf{m} \in \mathbb{Z}^d$ are straightforward to apply, and can be represented via DDNM's linear degeneration matrix $\mathbf{A}$ (32). However, modern generators using VAEs $\{\mathcal{E}, \mathcal{D}\}$ create compressed latent representations where direct masking becomes non-trivial due to lost spatial correspondence. Simple approaches like resizing pixel masks to latent space lose accuracy, while decoding to pixel space introduces computational overhead and potential information loss.

We propose *latent mask fuser* $\mathcal{M}(\cdot)$ to achieve fast video masking in video latent space. The module leverages a light-weight convolutional architecture. Let $\tilde{\mathbf{m}} \in \mathbb{Z}^l$ denote the mask in pixel space with length $l = T \times H \times W \times 3$, where the capital letters $T, H, W$ are the video dimensions in pixel space. Given two videos in the latent space $\mathbf{x}_1 \in \mathbb{R}^d, \mathbf{x}_2 \in \mathbb{R}^d$, the latent mask fuser computes the output via

$$\mathbf{x} = \mathcal{M}(\mathbf{x}_1, \mathbf{x}_2, \mathcal{E}(\tilde{\mathbf{m}})) \tag{15}$$

where $\mathcal{E}(\tilde{\mathbf{m}})$ is the encoded mask. With this, we can achieve fast and accurate video masking and replacement in latent space. Let $\tilde{\mathbf{x}}_1 \in \mathbb{R}^l, \tilde{\mathbf{x}}_2 \in \tilde{\mathbb{R}}^d$ denote the videos in pixel space and $\mathbf{x}_1 = \mathcal{E}(\tilde{\mathbf{x}}_1), \mathbf{x}_2 = \mathcal{E}(\tilde{\mathbf{x}}_2)$. We aim to approximate the masking operation in pixel space

$$\tilde{\mathbf{x}} = \{\tilde{\mathbf{x}}_1, \tilde{\mathbf{x}}_2\}, \quad \tilde{\mathbf{x}}^i = \begin{cases} \tilde{\mathbf{x}}_1^i, & \text{if } \tilde{\mathbf{m}}_i = 1 \\ \tilde{\mathbf{x}}_2^i, & \text{if } \tilde{\mathbf{m}}_i = 0 \end{cases}, \tag{16}$$

where we use the superscript $i$ to denote the element at the $i$-th entry. Then the training loss for the latent mask fuser is

$$\mathcal{L} = \mathbb{E}_{\tilde{\mathbf{x}}_1, \tilde{\mathbf{x}}_2, \tilde{\mathbf{m}}} \left[\|\tilde{\mathbf{x}} - \mathbf{x}\| + \lambda \cdot \text{LPIPS}(\tilde{\mathbf{x}}, \mathbf{x})\right]. \tag{17}$$

we use a series of simplex noise generators to produce dynamic random masks $\tilde{\mathbf{m}}$.

## 4 Experiments

### 4.1 Details

We apply our method to a video diffusion models CogVideoX (13; 38), which uses a diffusion transformer architecture. We also show ZeroPatcher can be used in HunyuanVideo, please see it in

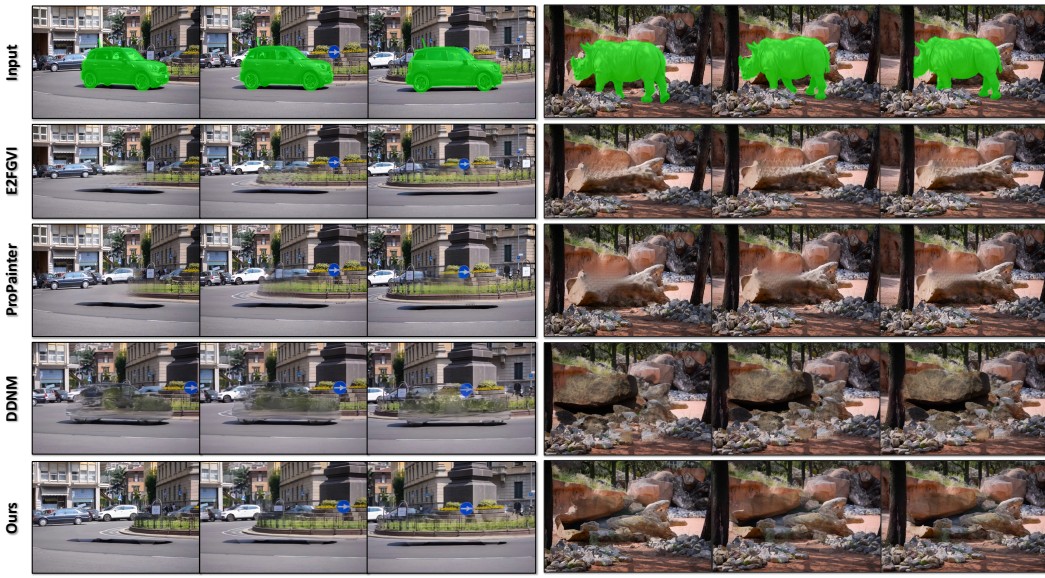

Figure 4: A visual comparison in video inpainting.

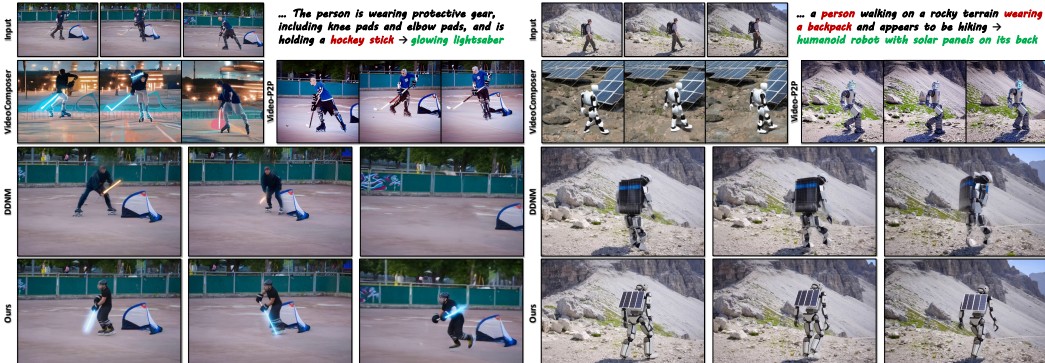

Figure 5: A visual comparison in video editing. We mark the changes in editing textual prompt using red (source) and green (editing).

the appendix. With CogVideoX, we generate $49 \times 720 \times 480$ videos using discrete euler sampler (11). To use dynamic video mask, we train the latent mask fuser on CogVideoX Video VAE using a small subset including 20,000 smaples from Panda-70M(5). The video resolution used during the training is $17 \times 256 \times 256$. The exact inference hyperparameters, the training configuration of the latent mask fuser, and a comprehensive computation and memory cost analysis can be found in the appendix.

For video inpainting evaluation, we conduct experiments on DAVIS (22) (50 videos) and YouTube-VOS (35) (508 videos) using their original splits (44). As CogVideoX requires text guidance, we use LLaVa-NeXt (41) to generate precise original prompts identifying main objects, then modify them with GPT-4 mini o1 by removing object descriptions to create inpainting prompts. Following (44), we evaluate using PSNR and SSIM for unmasked region similarity to source videos, VFID for visual realism, and $E_{\mathrm{warp}}$ for temporal consistency, with all inputs center-cropped to model resolution.

For video editing evaluation, we test on 50 DAVIS videos following (42; 18), using GPT-4 mini o1 to create editing prompts by modifying original video descriptions. We assess performance using CLIP scores for text alignment, PSNR/SSIM for background preservation, and $E_{\mathrm{warp}}$ for temporal smoothness. Additionally, we conduct human evaluation collecting 1,310 responses from 171 participants who compared results on DAVIS.

| Model | Youtube-VOS | | | | DAVIS | | | |
|---|---|---|---|---|---|---|---|---|
| | PSNR↑ | SSIM ↑ | VFID↓ | $E_{\mathrm{warp}} \downarrow$ | PSNR↑ | SSIM ↑ | VFID↓ | $E_{\mathrm{warp}} \downarrow$ |
| Trained | | | | | | | | |
| FuseFormer | 33.32 | 0.9681 | 0.053 | 1.053 | 32.59 | 0.9701 | 0.137 | 1.349 |
| ISVI | 30.34 | 0.9458 | 0.077 | 1.008 | 32.17 | 0.9588 | 0.189 | 1.291 |
| FGT | 32.17 | 0.9599 | 0.054 | 1.025 | 32.86 | 0.9650 | 0.129 | 1.323 |
| E2FGVI | 33.71 | 0.9700 | 0.046 | 1.013 | 33.01 | 0.9721 | 0.116 | 1.289 |
| ProPainter | **34.43** | **0.9735** | **0.042** | **0.974** | **34.47** | **0.9776** | **0.098** | **1.187** |
| Training-free | | | | | | | | |
| SDEdit | 26.41 | 0.8436 | 0.089 | 0.988 | 25.89 | 0.8266 | 0.203 | 1.266 |
| DDNM | 34.13 | 0.9705 | 0.067 | 1.069 | 33.97 | 0.9744 | 0.127 | 1.376 |
| Ours | **34.33** | **0.9715** | **0.039** | **0.961** | **34.13** | **0.9765** | **0.096** | **1.163** |

Table 1: Quantitative comparisons of video inpainting on Youtube-VOS (36) and DAVIS (22). Note that we ensure the number of computation using the denoising network are 100 for all training-free methods for a fair comparison. This indicates the computation cost for SDEdit, DDNM, and ours are almost the same in this experiment.

| Model | Automatic metrics | | | | Human evaluation | |
|---|---|---|---|---|---|---|
| | CLIP score | PSNR | SSIM | $E_{\mathrm{warp}}$ | BC | TA |
| PF | 0.3341 | 34.07 | 0.9684 | 1.513 | 30.6% | 2.3 % |
| VideoComposer | 0.3172 | - | - | 1.361 | - | 12.8 % |
| Video-P2P | 0.3313 | 30.47 | 0.9433 | 1.457 | 9.3% | 15.3 % |
| DDNM | 0.3267 | 33.48 | 0.9649 | 1.264 | 29.2% | 8.9 % |
| Ours | **0.3543** | **34.23** | **0.9712** | **1.231** | **30.9%** | **60.7%** |

Table 2: Quantitative comparisons of editing performance. PF denotes per-frame video inpainting using stable diffusion inpainting (24). BC represents back ground preservation. TA denotes text-video alignment. We ignore PSNR, SSIM, and TA metrics in VideoComposer because it does not have the ability to maintain the background.

## 4.2 Comparisons

We evaluate ZeroPatcher on both video inpainting and editing tasks, comparing against training-free baselines and trained models.

**Computation and memory cost** We show the throughput and computation overhead of ZeroPatcher in the appendix.

**Video inpainting** We compare against trained methods (FuseFormer (17), ISVI (40), FGT (39), E2FGVI (16), ProPainter (44)) and training-free baselines (SDEdit (20), DDNM (32)). To use SDEdit in video inpainting, we fill the main object with the mean colour of the video background. For a fair comparison, the number of function evaluations (NFE) is 100 for all training-free methods. The results are presented in table 1. ZeroPatcher achieves the state-of-the-art performance within the training-free methods and is very competitive even compared with the training based counterparts. Note that we achieved the best performance in $E_{\mathrm{warp}}$ in Youtube-VOS among all the methods. A visual comparison is provided in fig. 4. Our method has strong generation capability and can preserve good consistency between the removed parts and the background. The generation ability is very useful especially when the inpainting region is large and lack of references.

**Video editing** We compare ZeroPatcher with stable diffusion inpainting model (24), VideoComposer (30), Video-P2P (18) and DDNM (32). We rephrase our editing text prompts for Video-P2P, because it only allows users to change a word from the original description. The quantitative results are shown in table 2. ZeroPather achieves the state-of-the-art performance in quantitative metrics. Notably, VideoComposer uses depth to guide video editing, so we ignore the background similarity

| Model | PSNR↑ | SSIM ↑ | VFID↓ | $E_{\text{warp}} \downarrow$ |
|---|---|---|---|---|
| Ours | 34.13 | 0.9765 | 0.106 | 1.213 |
| w.o. CD-EM | 33.89 | 0.9711 | 0.123 | 1.355 |
| w.o. LMF | 33.61 | 0.9673 | 0.125 | 1.361 |
| w.o. BP | 17.41 | 0.5351 | 0.116 | 1.224 |

Table 3: Ablation results on the technical components. We use BP to represent back projection and use LMF to represent the latent mask fuser.

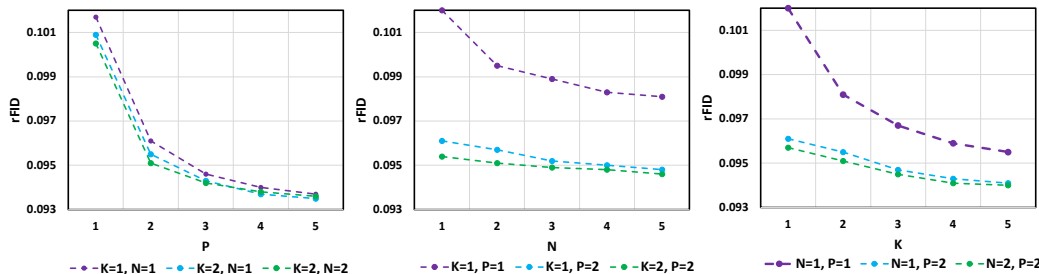

Figure 6: Ablation experiment on choice of $P$, $N$, and $K$ parameters.

for VideoComposer. A visual comparison is shown in fig. 5. Our method achieves the best text-video alignment, video smoothness, and background similarity.

### 4.3 Ablation

We evaluated the individual technical components of ZeroPatcher to study their effectiveness. Specifically, the ablation experiments are finished based on the pretrained CogVideoX (38) generator to produce $49 \times 720 \times 480$ videos. We measure the impact of technical components including CD-EM, back projection, and latent mask fuser. The results are shown in table 3. Using CD-EM or back projection alone cannot achieve an ideal performance. Incorporating both methods can exploit the capability of the foundation model to the limit. When we are using dynamic video masks, the latent mask fuser improves a lot in terms of visual fidelity compared with using resized masks. Meanwhile, the visual comparison shows our method achieves accurate video masking with subtle reconstruction loss.

A series of ablation experiments are conducted to validate the parameters in ZeroPatcher. We make a grid search over the sample number of Monte-Carlo integration $N$, the fixed point iteration step number $K$, and the EM outer-loop iteration number $P$. The results are shown in fig. 6. We perform video inpainting on DAVIS and compute VFID. The experiment suggests when $N$, $K$, and $P$ become larger, the performance of our method increases monotonically. Compared to the case $N = 1, K = 1, P = 2$, the largest computation setting achieves approximately 3% improvement on VFID. The iteration speed is shown in the appendix. Our method is able to achieve convincing performance without considerably more computation.

## 5   Conclusion and limitation

In this paper, we present *ZeroPatcher*, a training-free plugin to introduce video inpainting and editing capability for pre-trained video foundation models. Our method optimizes denoising trajectory given the unmasked video through *conditional denoising expectation maximization*. To achieve the video masking in video VAE features, we propose latent mask fuser to achieve fast and accurate video masked fusing. ZeroPatcher is widely applicable to prevalent diffusion generators and demonstrates satisfactory performance in inpainting and editing tasks. ZeroPatcher relies on the prior knowledge of a pretrained text-to-video generation model to perform video inpainting and editing. As a result, if the input data or textual prompts are beyond the domain of the pretrained model, our method may not be able to correctly render the video.

**Acknowledgement**   This work was supported by Alibaba Innovative Research Program.

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

# A    Detailed deduction of the fixed point method

Below, we show the matrix formation for the fixed point iteration that soles CD-EM. To simplify the equation, we use two constants $c_{t-1}^1, c_{t-1}^2$ to represent the coefficients. $c_{t-1}^1$ and $c_{t-1}^2$ are mathematically

$$c_{t-1}^1 = \sqrt{\frac{1 - \alpha_{t-1} - \sigma_{t-1}^2}{1 - \alpha_t}}, \quad c_{t-1}^2 = \frac{\sqrt{\alpha_{t-1}(1 - \alpha_t)} - \sqrt{1 - \alpha_{t-1} - \sigma_{t-1}^2} \cdot \sqrt{\alpha_t}}{\sqrt{1 - \alpha_t}}. \tag{18}$$

We aim to make the equation holds:

$$(\mathbf{x}_{t-1}^a, \mathbf{x}_{t-1}^{b,i}) - \boldsymbol{\mu}_{\boldsymbol{\theta}}(\mathbf{x}_t) = (\mathbf{x}_{t-1}^a, \mathbf{x}_{t-1}^{b,i}) + c_{t-1}^1 \mathbf{x}_t + c_{t-1}^2 \mathbf{f}_{\boldsymbol{\theta}}(\mathbf{x}_t). \tag{19}$$

We aim to find a $\mathbf{x}_t$ that ensures

$$(\mathbf{x}_{t-1}^a, \mathbf{x}_{t-1}^{b,i}) + c_{t-1}^1 \mathbf{x}_t + c_{t-1}^2 \mathbf{f}_{\boldsymbol{\theta}}(\mathbf{x}_t) = 0 \tag{20}$$

in order to maximize the expectation log-likelihood. Unluckily, this equation can not be solved analytically since $\mathbf{f}_{\boldsymbol{\theta}}(\cdot)$ is a neural function. In alternative, we propose to use fixed point iteration to get an approximated solution. We rewrite eq. (20) to matrix formation. Mathmatically that is

$$- \begin{bmatrix} ((\mathbf{x}_{t-1}^a, \mathbf{x}_{t-1}^{b,1}) + c_{t-1}^2 \mathbf{f}_{\boldsymbol{\theta}}(\mathbf{x}_t')); \\ ((\mathbf{x}_{t-1}^a, \mathbf{x}_{t-1}^{b,2}) + c_{t-1}^2 \mathbf{f}_{\boldsymbol{\theta}}(\mathbf{x}_t')); \\ \cdots \\ ((\mathbf{x}_{t-1}^a, \mathbf{x}_{t-1}^{b,N}) + c_{t-1}^2 \mathbf{f}_{\boldsymbol{\theta}}(\mathbf{x}_t')) \end{bmatrix} = c_{t-1}^1 \begin{bmatrix} \mathbf{I} \\ \mathbf{I} \\ \cdots \\ \mathbf{I} \end{bmatrix} \mathbf{x}_t \tag{21}$$

where we use ";" to denote concatenation in column, and $\mathbf{x}_t'$ is $\mathbf{x}_t$ at the previous fixed point iteration step. We use $\mathbb{I}$ to denote the $Nd \times d$ matrix $\mathbb{I} = [\mathbf{I} \quad \mathbf{I} \quad \cdots \quad \mathbf{I}]^T$. The fixed point iteration follows a formation of least square solution. Specifically, the solution is

$$\mathbf{x}_t \leftarrow -\frac{1}{c_{t-1}^1} (\mathbb{I}^T \mathbb{I})^{-1} \mathbb{I} \begin{bmatrix} ((\mathbf{x}_{t-1}^a, \mathbf{x}_{t-1}^{b,1}) + c_{t-1}^2 \mathbf{f}_{\boldsymbol{\theta}}(\mathbf{x}_t')); \\ ((\mathbf{x}_{t-1}^a, \mathbf{x}_{t-1}^{b,2}) + c_{t-1}^2 \mathbf{f}_{\boldsymbol{\theta}}(\mathbf{x}_t')); \\ \cdots \\ ((\mathbf{x}_{t-1}^a, \mathbf{x}_{t-1}^{b,N}) + c_{t-1}^2 \mathbf{f}_{\boldsymbol{\theta}}(\mathbf{x}_t')) \end{bmatrix}. \tag{22}$$

# B    Proof of theorem 3.1

We show the proof in this section.

*Proof.* We first study what we need $\mathbf{f}_{\theta}(\cdot)$ to be like in order to ensure an unique fixed point. We use $g(\cdot)$ to denote the iteration step

$$\mathbf{x}_t \leftarrow g(\mathbf{x_t}), \quad g(\mathbf{x}_t) = -\frac{(\mathbf{x}_{t-1}^a, \mathbf{x}_{t-1}^{b,1}) + c_{t-1}^2 \mathbf{f}_{\theta}(\mathbf{x}_t)}{c_{t-1}^1}. \tag{23}$$

According to *Banach's fixed point theorem*, we need $g(\cdot)$ to guarantee smaller than 1 Lipschitz constant

$$\|g(\mathbf{x}_t^1) - g(\mathbf{x}_t^2)\|_2 < 1 \cdot \|\mathbf{x}_t^1 - \mathbf{x}_t^2\|, \tag{24}$$

for arbitrary $\mathbf{x}_t^1, \mathbf{x}_t^2$. Expanding $g(\cdot)$ to the $\mathbf{f}_{\theta}(\cdot)$ formation, we derive

$$\left\| -\frac{(\mathbf{x}_{t-1}^a, \mathbf{x}_{t-1}^{b,1}) + c_{t-1}^2 \mathbf{f}_{\theta}(\mathbf{x}_t^1)}{c_{t-1}^1} + \frac{(\mathbf{x}_{t-1}^a, \mathbf{x}_{t-1}^{b,1}) + c_{t-1}^2 \mathbf{f}_{\theta}(\mathbf{x}_t^2)}{c_{t-1}^1}. \right\|_2 < \|\mathbf{x}_t^1 - \mathbf{x}_t^2\|_2 \tag{25}$$

$$\iff \frac{c_{t-1}^2}{c_{t-1}^1} \|\mathbf{f}_{\theta}(\mathbf{x}_t^1) - \mathbf{f}_{\theta}(\mathbf{x}_t^2)\|_2 < \|\mathbf{x}_t^1 - \mathbf{x}_t^2\|_2 \tag{26}$$

$$\iff \|\mathbf{f}_{\theta}(\mathbf{x}_t^1) - \mathbf{f}_{\theta}(\mathbf{x}_t^2)\|_2 < \frac{c_{t-1}^1}{c_{t-1}^2} \|\mathbf{x}_t^1 - \mathbf{x}_t^2\|_2. \tag{27}$$

Therefore, we require the Lipschitz constant of $\mathbf{f}_\theta(\cdot)$ denoted by $\mathcal{L}_f$ to meet $\mathcal{L}_f < \frac{c_{t-1}^1}{c_{t-1}^2}$. Note that $c_{t-1}^1 > 0$ and $c_{t-1}^2 > 0$. The former one is obvious since we pre-define $\sigma_{t-1}$ to satisfy $1 - \alpha_{t-1} - \sigma_{t-1}^2 > 0$. In a diffusion model, $\alpha_t < \alpha_{t-1}, t \in \{1, \cdots, T\}$. Therefore we also derive $\sqrt{1 - \alpha_t} > \sqrt{1 - \alpha_{t-1} - \sigma_{t-1}^2}$ which makes the denominator of $c_{t-1}^2$ positive. We then relate the Lipschitz constant $\mathcal{L}_f$ to that of the real score function $\mathcal{L}_s$. The clean data estimant can be expressed by

$$\mathbf{f}_\theta(\mathbf{x}_t) = \frac{\mathbf{x}_t - \sqrt{1 - \alpha_t}\epsilon_\theta(\mathbf{x}_t)}{\sqrt{\alpha_t}} = \frac{\mathbf{x}_t + \mathbf{s}_\theta(\mathbf{x}_t)}{\sqrt{\alpha_t}}. \tag{28}$$

We construct an inequality that can be the sufficient condition for to find the Lipschitz constant of $\mathbf{f}_\theta(\cdot)$. Assume the following inequality is satisfied

$$1 + \mathcal{L}_s < \frac{c_{t-1}^1 \sqrt{\alpha_t}}{c_{t-1}^2} \tag{29}$$

$$\Longleftrightarrow \|\mathbf{x}_t^1 - \mathbf{x}_t^2\|_2 + \mathcal{L}_s\|\mathbf{x}_t^1 - \mathbf{x}_t^2\|_2 < \frac{c_{t-1}^1 \sqrt{\alpha_t}}{c_{t-1}^2}\|\mathbf{x}_t^1 - \mathbf{x}_t^2\|_2 \tag{30}$$

$$\overset{\text{eq. (12)}}{\Longleftrightarrow} \|\mathbf{x}_t^1 - \mathbf{x}_t^2\|_2 + \|\mathbf{s}_\theta(\mathbf{x}_t^1) - \mathbf{s}_\theta(\mathbf{x}_t^2)\|_2 < \|\mathbf{x}_t^1 - \mathbf{x}_t^2\|_2 + \mathcal{L}_s\|\mathbf{x}_t^1 - \mathbf{x}_t^2\|_2 < \frac{c_{t-1}^1 \sqrt{\alpha_t}}{c_{t-1}^2}\|\mathbf{x}_t^1 - \mathbf{x}_t^2\|_2 \tag{31}$$

$$\overset{\text{triangular}}{\underset{\text{inequality}}{\Longleftrightarrow}} \|(\mathbf{x}_t^1 - \mathbf{x}_t^2) + (\mathbf{s}_\theta(\mathbf{x}_t^1) - \mathbf{s}_\theta(\mathbf{x}_t^2))\|_2 <$$

$$\|\mathbf{x}_t^1 - \mathbf{x}_t^2\|_2 + \|\mathbf{s}_\theta(\mathbf{x}_t^1) - \mathbf{s}_\theta(\mathbf{x}_t^2)\|_2 < \frac{c_{t-1}^1 \sqrt{\alpha_t}}{c_{t-1}^2}\|\mathbf{x}_t^1 - \mathbf{x}_t^2\|_2 \tag{32}$$

$$\Longleftrightarrow \left\|\frac{\mathbf{x}_t^1 + \mathbf{s}_\theta(\mathbf{x}_t^1)}{\sqrt{\alpha_t}} - \frac{\mathbf{x}_t^2 + \mathbf{s}_\theta(\mathbf{x}_t^2)}{\sqrt{\alpha_t}}\right\|_2 = \|\mathbf{f}_\theta(\mathbf{x}_t^1) - \mathbf{f}_\theta(\mathbf{x}_t^2)\|_2 < \frac{c_{t-1}^1}{c_{t-1}^2}\|\mathbf{x}_t^1 - \mathbf{x}_t^2\|_2. \tag{33}$$

Therefore, if $1 + \mathcal{L}_s < \frac{c_{t-1}^1 \sqrt{\alpha_t}}{c_{t-1}^2}$ is satisfied, we ensure an unique fixed point can be found. Then, the Lipschitz constant of the real score function $\mathbf{s}(\cdot)$ is computed below. The score function is defined as

$$\mathbf{s}(\mathbf{x}_t|\mathbf{x}_0) \doteq \nabla_{\mathbf{x}_t} \log p(\mathbf{x}_t|\mathbf{x}_0) = -\frac{\mathbf{x}_t - \sqrt{\alpha_t}\mathbf{x}_0}{1 - \alpha_t}. \tag{34}$$

We derive the Lipschitz constant through

$$\|\mathbf{s}(\mathbf{x}_t^1|\mathbf{x}_0) - \mathbf{s}(\mathbf{x}_t^2|\mathbf{x}_0)\|_2 = \left\|-\frac{\mathbf{x}_t^1 - \sqrt{\alpha_t}\mathbf{x}_0}{1 - \alpha_t} + \frac{\mathbf{x}_t^2 - \sqrt{\alpha_t}\mathbf{x}_0}{1 - \alpha_t}\right\|_2 \tag{35}$$

$$= \frac{1}{1 - \alpha_t}\|\mathbf{x}_t^1 - \mathbf{x}_t^2\|_2 < \mathcal{L}_s\|\mathbf{x}_t^1 - \mathbf{x}_t^2\|_2. \tag{36}$$

Therefore, the Lipscthiz constant is $\mathcal{L}_s = 1/(1 - \alpha_t)$. Replacing this and the definition of $c_{t-1}^1, c_{t-1}^2$ to eq. (29) we have

$$\frac{2 - \alpha_t}{1 - \alpha_t} < \frac{\sqrt{(1 - \alpha_{t-1} - \sigma_{t-1}^2)\alpha_t}}{\sqrt{\alpha_{t-1}(1 - \alpha_t)} - \sqrt{(1 - \alpha_{t-1} - \sigma_{t-1}^2)\alpha_t}} \tag{37}$$

$$\overset{\text{rearrange}}{\Longleftarrow} (2 - \alpha_t)\left(\sqrt{\alpha_{t-1}(1 - \alpha_t)} - \sqrt{(1 - \alpha_{t-1} - \sigma_{t-1}^2)\alpha_t}\right) < (1 - \alpha_t)\sqrt{(1 - \alpha_{t-1} - \sigma_{t-1}^2)\alpha_t} \tag{38}$$

$$\Longleftrightarrow (2 - \alpha_t)\sqrt{\alpha_{t-1}(1 - \alpha_t)} < (3 - 2\alpha_t)\sqrt{\alpha_t(1 - \alpha_{t-1} - \sigma_{t-1}^2)} \tag{39}$$

Knowing whether this inequality holds is not obvious. Instead of seeking what kind of noise schedule makes the condition holds, we study the commonly used noise schedules. We use $m(t) =$

**Algorithm 1:** Conditional denoising expectation maximization for diffusion model

1 **Input**: Known video segment $\boldsymbol{x}_0^a$, pre-trained diffusion model $\boldsymbol{f}_\theta(\cdot)$, stopping criteria $\tau$, inference timesteps $T$, EM iteration number $P$, Monte-Carlo sample number $N$, fixed point iteration step number $K$
2 **Output**: Rendered video segment $\boldsymbol{x}_0^b$
3 $\boldsymbol{x}_T \sim \mathcal{N}(\mathbf{0}, \mathbf{I})$
4 **for** $t = T$ **to** $0$ **do**
5    **for** $p = 1$ **to** $P$ **do**
6       $\boldsymbol{x}_t^{\text{old}} \leftarrow \boldsymbol{x}_t$
7       **for** $i = 1$ **to** $N$ **do**
8          $\boldsymbol{x}_{t-1}^i \sim p_\theta(\boldsymbol{x}_{t-1}^b | \boldsymbol{x}_t^{\text{old}})$
9       **end**
10       $\epsilon \sim \mathcal{N}(\mathbf{0}, \mathbf{I})$
11       $\boldsymbol{x}_{t-1}^a \leftarrow \sqrt{\alpha_{t-1}}\boldsymbol{x}_0^a + \sqrt{1 - \alpha_{t-1}}\epsilon$
12       **for** $k = 1$ **to** $K$ **do**
13          Apply eq. (22) to get $\boldsymbol{x}_t$
14       **end**
15    **end**
16    $\boldsymbol{x}_{t-1} \sim p_\theta(\boldsymbol{x}_{t-1} | \boldsymbol{x}_t)$
17 **end**
18 **return** $\boldsymbol{x}_0$

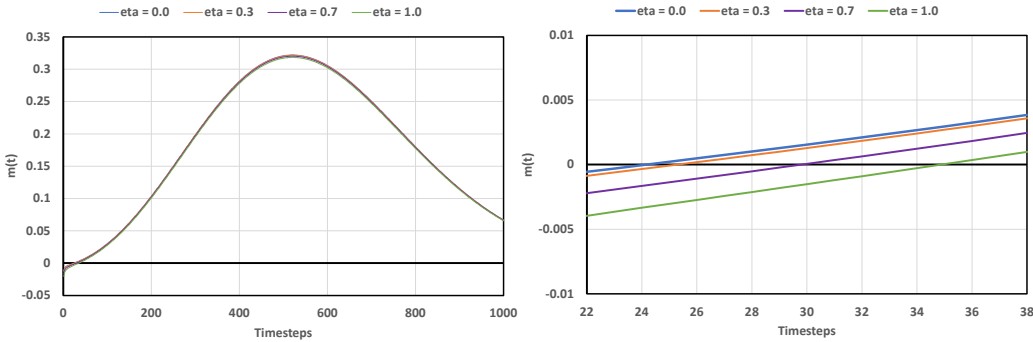

Figure 7: The curve of $m(t)$ under different settings of $\eta$, The right curve zoom in the regions that curves hit zero line.

$(3-2\alpha_t)\sqrt{\alpha_t(1 - \alpha_{t-1} - \sigma_{t-1}^2)} - (2-\alpha_t)\sqrt{\alpha_{t-1}(1 - \alpha_t)}$ for simplification. Then the convergency and uniqueness hold when $m(t) > 0$. We plot $m(t)$ curve for a series of diffusion schedules. The results are shown in fig. 7. We plot the situation of CogVideoX scheduler with different choices of $\eta$. In most sampling timestep, we can ensure uniqueness and convergency. □

## C  CD-EM on flow matching generators

Our method is not only applicable to diffusion based generators. Instead, CD-EM is also applicable to flow matching generative models. Consider a model that uses the following flows and velocity field in $t \in [0, T)$.

$$\mathbf{x}_t = \phi_t(\mathbf{x}_0) = \sigma(t)\mathbf{x}_0 + (1 - \sigma(t))\epsilon, \quad u_t(\phi_t(\mathbf{x}_0)) = \frac{\partial}{\partial t}\phi_t(\mathbf{x}_0) = \frac{\partial \sigma(t)}{\partial t} \cdot (\mathbf{x}_0 - \epsilon), \quad (40)$$

where $\sigma(\cdot)$ is the noise schedule, and $\epsilon$ is a random gaussian noise. The velocity field $u_t(\cdot)$ will support a probability path

$$q(\mathbf{x}_t | \mathbf{x}_0) = \mathcal{N}(\mathbf{x}_t | \sigma(t)\mathbf{x}_0, (1 - \sigma(t))^2\mathbf{I}). \quad (41)$$

Solving the ODE in Eq.(4.1.1) produces the sampling result. Euler sampler is commonly used in which the iteration is

$$\mathbf{x}_s = \mathbf{x}_t + (s - t)u_t(\mathbf{x}_t), \quad 0 \le s < t \le T \tag{42}$$

To optimize the model throught flow matching, one usually use a neural network $\mathbf{v}_\theta(\cdot)$ to approximate the scaled velocity field $\mathbf{v}_\theta(\mathbf{x}_t) \simeq \epsilon - \mathbf{x}_0, u_t(\mathbf{x}_t) \simeq -\partial\sigma(t)/\partial t \cdot \mathbf{v}_\theta(\mathbf{x}_t)$. Replace the velocity in Eq.(4.1.3) with the model estimation, we derive

$$\mathbf{x}_s = \mathbf{x}_t + (\sigma(t) - \sigma(s))\mathbf{v}_\theta(\mathbf{x}_t). \tag{43}$$

To optimize the trajectory with expectation maximization method, we need to know the sampling posterior distribution $q(\mathbf{x}_s|\mathbf{x}_t, \mathbf{x}_0), 0 \le s < t \le T$ of the flow model. Applying *bayes formula*, we get

$$q(\mathbf{x}_s|\mathbf{x}_t, \mathbf{x}_0) = \frac{q(\mathbf{x}_t|\mathbf{x}_s, \mathbf{x}_0)q(\mathbf{x}_s|\mathbf{x}_0)}{q(\mathbf{x}_t|\mathbf{x}_0)}. \tag{44}$$

The distribution $q(\mathbf{x}_s|\mathbf{x}_0)$ and $q(\mathbf{x}_t|\mathbf{x}_0)$ are already known. However, since a flow model is deterministic, the term in the nominator $q(\mathbf{x}_t|\mathbf{x}_s, \mathbf{x}_0)$ is specifically Dirac delta. Therefore, the sampling distribution will also be Dirac delta that

$$q(\mathbf{x}_s|\mathbf{x}_t, \mathbf{x}_0) = \delta\left(\mathbf{x}_s - \frac{1 - \sigma(s)}{1 - \sigma(t)} \cdot \mathbf{x}_t + \frac{\sigma(t) - \sigma(s)}{1 - \sigma(t)} \cdot \mathbf{x}_0\right). \tag{45}$$

Meanwhile, we also derive the the modeled sampling distribution

$$p_\theta(\mathbf{x}_s|\mathbf{x}_t) = \delta\left(\mathbf{x}_s - \mathbf{x}_t - (\sigma(t) - \sigma(s))\mathbf{v}_\theta(\mathbf{x}_t)\right). \tag{46}$$

Given this, we are able to maximize the expectation in Eq.(2.2.3)

$$Q(\mathbf{x}_t|\mathbf{x}_t^{\mathrm{old}}) \simeq \sum_i^N \log p_\theta(\mathbf{x}_s^a, \mathbf{x}_s^{b,i}|\mathbf{x}_t)d\mathbf{x}_s^b, \tag{47}$$

which requires $p_\theta(\mathbf{x}_s^a, \mathbf{x}_s^{b,i}|\mathbf{x}_t)$ to be maximized. In this case, we expect the following equation set to hold

$$\begin{cases} (\mathbf{x}_s^a, \mathbf{x}_s^{b,1}) = \mathbf{x}_t + (\sigma(t) - \sigma(s))\mathbf{v}_\theta(\mathbf{x}_t^{\mathrm{old}}) \\ (\mathbf{x}_s^a, \mathbf{x}_s^{b,2}) = \mathbf{x}_t + (\sigma(t) - \sigma(s))\mathbf{v}_\theta(\mathbf{x}_t^{\mathrm{old}}) \\ \cdots \\ (\mathbf{x}_s^a, \mathbf{x}_s^{b,N}) = \mathbf{x}_t + (\sigma(t) - \sigma(s))\mathbf{v}_\theta(\mathbf{x}_t^{\mathrm{old}}) \end{cases} \Longleftrightarrow \begin{bmatrix} (\mathbf{x}_s^a, \mathbf{x}_s^{b,1}) + (\sigma(s) - \sigma(t))\mathbf{v}_\theta(\mathbf{x}_t^{\mathrm{old}}); \\ (\mathbf{x}_s^a, \mathbf{x}_s^{b,2}) + (\sigma(s) - \sigma(t))\mathbf{v}_\theta(\mathbf{x}_t^{\mathrm{old}}); \\ \cdots \\ (\mathbf{x}_s^a, \mathbf{x}_s^{b,N}) + (\sigma(s) - \sigma(t))\mathbf{v}_\theta(\mathbf{x}_t^{\mathrm{old}}) \end{bmatrix} = \begin{bmatrix} \mathbf{I} \\ \mathbf{I} \\ \cdots \\ \mathbf{I} \end{bmatrix} \mathbf{x}_t. \tag{48}$$

$\mathbf{I} \in \mathbb{R}^{n \times n}$ is an identity matrix. We solve this equation set with least square and produces the fixed point iteration step

$$\mathbf{x}_t \leftarrow (\mathbb{I}^T\mathbb{I})^{-1}\mathbb{I} \begin{bmatrix} (\mathbf{x}_s^a, \mathbf{x}_s^{b,1}) + (\sigma(s) - \sigma(t))\mathbf{v}_\theta(\mathbf{x}_t^{\mathrm{old}}); \\ (\mathbf{x}_s^a, \mathbf{x}_s^{b,2}) + (\sigma(s) - \sigma(t))\mathbf{v}_\theta(\mathbf{x}_t^{\mathrm{old}}); \\ \cdots \\ (\mathbf{x}_s^a, \mathbf{x}_s^{b,N}) + (\sigma(s) - \sigma(t))\mathbf{v}_\theta(\mathbf{x}_t^{\mathrm{old}}) \end{bmatrix}, \tag{49}$$

where we use $\mathbb{I}$ to denote the $Nd \times d$ matrix $\mathbb{I} = \begin{bmatrix} \mathbf{I} & \mathbf{I} & \cdots & \mathbf{I} \end{bmatrix}^T$. Similarly, when $N = 1$, the iteration step is implified to

$$\mathbf{x}_t \leftarrow (\mathbf{x}_s^a, \mathbf{x}_s^b) + (\sigma(s) - \sigma(t))\mathbf{v}_\theta(\mathbf{x}_t^{\mathrm{old}}). \tag{50}$$

To uniqueness and convergency of the fixed point iteration can be found using a similar way in appendix B. We try this method on a flow matching video generator named HunyuanVideo (33) to perform video editing. Please see our supplementary material.

## D   Computation and memory cost

**Ablation experiment**   The computation and memory cost of ZeroPathcer is shown in table 4. The theoretical complexity considers the computation required for each denoising step. We let the complexity for every function evaluation be $O(1)$. $P$ is the outer loop step number of the expectation maximization. $N$ denotes how many $\mathbf{x}_{t-1}^b$ are sampled in order to compute the likelihood expectation. $K$ is the step number of the fixed point iteration. With a reasonable set of hyperparameters (i.e. $P = 2, N = 1, K = 1$), ZeroPathcer will not introduce enormous computation compared to the direct text-to-video inference. Meanwhile, we show that the ZeroPathcer does not use considerable addition memory which is usually required for attention manipulation methods like Video-P2P (18) and FateZero (23).

| Model | Theoretical complexity | Memory | Speed |
|---|---|---|---|
| Inference params | $P = 2, N = 1, K = 1$ | | |
| t2v model | $O(1)$ | 22,489MB | 2.61 s/it |
| + BP | $O(1)$ | 22.512MB | 2.63 s/it |
| + CD-EM | $O(1 + PN + P(K-1))$ | 22,567MB | 7.41 s/it |
| + LMF | $O(1 + PN + P(K-1))$ | 22.893MB | 7.63 s/it |

Table 4: Theoretical and practical computation cost analysis of our method. Theoretical complexity and speed show the computation required for each denoising step. We use LMF to represent the latent mask fuser. The speed is computed under seconds per iteration.

| Method | Resolution | Memory | NFE | Speed |
|---|---|---|---|---|
| Text-to-video | | | | |
| CogVideoX | $49 \times 720 \times 480$ | 22.488MB | 30 | 78.3 s/sample |
| Inpainting | | | | |
| SDEdit | $49 \times 720 \times 480$ | 22.503MB | 100 | 240.3 s/sample |
| DDNM | $49 \times 720 \times 480$ | 22.512MB | 100 | 241.7 s/sample |
| Ours | $49 \times 720 \times 480$ | 22,567MB | 100 | 250.9 s/sample |
| Editing | | | | |
| PF | $8 \times 512 \times 512$ | 4,279 MB | 50 | 81.2 s/sample |
| VideoComposer | $16 \times 256 \times 256$ | 7,394MB | 50 | 101.4 s/sample |
| VideoP2P | $8 \times 512 \times 512$ | 19,453MB | 100 | 137.8 s/sample |
| DDNM | $49 \times 720 \times 480$ | 22.522MB | 100 | 241.5 s/sample |
| Ours | $49 \times 720 \times 480$ | 22,569MB | 100 | 249.4 s/sample |

Table 5: A comparison over computation and memory usage.

**Comparison with other methods**    We compare the memory and throughput among the methods. The result is shown in table 5. Our method achieves remarkable improvements over the training-free methods under similar computation cost. To achieve faster inference, one can choose to trade performance for speed by letting $N = 1, K = 1, P = 1$.

## E    Algorithmic description of CD-EM

We show an algorithmic description of CD-EM in Algo.1. It provides the clean CD-EM without back projection and latent mask fuser.

## F    Details of latent mask fuser

In this sector, we briefly introduce the architecture of our latent mask fuser. It is consists of 6 residual blocks, a convolutional input layer, and a convolutional output layer. After the 3-th block, we use an adaptive instance normalization layer (14) to perform mask fusing. Each residual block is made up of two convolution blocks (a concatenation of a GroupNorm layer, a SiLU layer, and a 3D convolution layer) and a skip connection layer achieved by a $1 \times 1 \times 1$ 3D convolution. During inference, we concatenate three latent features from the two source video and the mask in channel dimension, and then the feature is feed into the input layer. No upsampling and downsampling modules are used in latent mask fuser. The model uses a channel size of 1536.

## G    Inference details and hyperparameters

The inference hyperparameters are shown in table 6. We use DDIM diffusion sampler with sampling stochasticity factor $\eta = 1.0$. Our experiment shows using $\eta = 1.0$ can greatly increase video

| Hyperparameter | CogVideoX |
|---|---|
| EM outer loop steps $P$ | 2 |
| Number of samples in expectation $N$ | 1 |
| Fixed point iteration steps $K$ | 1 |
| Sampler | DDIM |
| Inference steps | 50 |
| CD-EM stop step | 25 |
| NFEs | 100 |
| Classifier-free guidance scale | 6 |
| eta | 1.0 |

Table 6: Hyperparameters use in inference.

consistency since the model will not follow the deterministic sampling trajectory. CD-EM is not always required in all denoising timesteps. We find using CD-EM in early sampling stages can already produce plausible results. Therefore, we add a stoping step for CD-EM at 25 to save computation without losing noticeable performance. With CD-EM hypermarameters set to $P = 2, N = 1, K = 1$, we ensure every denoising step with CD-EM will only cost 3 NFEs. With the remaining 25 steps using only 1 NFE, sampling through ZeroPatcher requires 100 NFEs in total. We use 8 NVIDIA A100 80G GPUs to run in parallel during inference to get results faster.

## H   Additional visual results

We attach result videos in our supplementary material. We would appreciate it if you check them.

## I   Broader impact

The integration of pre-trained text-to-video foundation models into video inpainting and editing workflows presents transformative opportunities alongside critical ethical challenges. By enabling high-quality video manipulation without additional training, this approach democratizes access to advanced editing tools, empowering independent creators and small studios to achieve professional-grade results—potentially revitalizing archival restoration efforts and lowering costs for cultural heritage preservation. However, the same capabilities raise significant concerns: the efficiency of dynamic object removal and latent-space masking could streamline the creation of convincing deepfakes, exacerbating misinformation risks in an era already plagued by synthetic media distrust. The model-agnostic nature of the method further amplifies these risks, as it could be applied to any foundation model, including those with fewer ethical safeguards.

