# OpenReview forum: "ZeroPatcher: Training-free Sampler for Video Inpainting and Editing"
_NeurIPS.cc/2025/Conference — NeurIPS 2025 poster_

### Official Review · Reviewer_4gUU · 2025-06-25

**Clarity:** 2
**Significance:** 3
**Originality:** 3
**Rating:** 4
**Confidence:** 4

**Summary:**

The authors propose a training-free video inpainting and editing method, named ZeroPatcher. The method is based on the back-projection operation of DDNM, and the authors claim that pure DDNM suffers from two key limitations: the estimation bias and not acquiring any information from previous time steps. Therefore, ZeroPatcher includes additional components such as CD-EM and LMF to reduce those side effects. The visual results reveal that ZeroPatcher outperforms recent methods, and can be extended to both flow-based and denoising-based architectures.

**Questions:**

1. RePaint and COPAINT use an analytical posterior + pixel-space masking that resembles ZeroPatcher’s EM + latent masking. Explain why EM is preferred.

2. I would assume one of the major advantages of CD-EM over latent optimization would be converging to a unique fixed point, as Theorem 3.1 claims. However, no visuals prove that increasing K and P leaves converged results unchanged, visually. I would suggest adding inpainting outputs alongside Figure 6 to further prove your point. Otherwise, why prefer EM over latent sample optimization?

3. Provide an explicit A pseudo-inverse example for a given masking matrix A and show how it is applied to $ x_0 $ via Equation 5.

4. It seems that the majority of the improvements are due to BP (i.e. DDNM), and the added components improve the quantitative metrics marginally. Include a visual ablation that matches Table 3, it might be easier to observe the improvements.

5. The latent spaces of such video models are well-aligned spatially with pixel domain (meaning that a pixel would be represented more or less on similar spatial regions in the latent space, across all latent channels, very unlike GANs' latent spaces). This means that resizing a pixel-space mask and applying it on the latent domain would more or less yield the desired results. However, the method indicates that performing latent masking would be better (Lines 186-187). Visually compare naive masking (pixel-space masking applied directly in latent space without LMF) with LMF.

6. In Figure 4 the car-roundabout example still shows visible shadows. Does extending the mask remove these?

These questions are related to Weaknesses #2. I stand my rating as borderline reject, but I am going to increase my score 1 to 2 points if these questions are adequately answered and the novelty is justified.

**Ethical Concerns:**

["NO or VERY MINOR ethics concerns only"]

**Final Justification:**

I find the rebuttal adequate, hence I increase my final score a point.

However, I expect to see the visual results of Q3-Q4-Q5, written explanations for Q1-Q2, and the tables provided in the rebuttal in the final paper & supplementary material. I believe that these are important for the sake of completeness.

**Limitations:**

There are no fail cases visualized as the limitations of the method. Also, see the weaknesses part.

**Paper Formatting Concerns:**

No formatting issues are observed.

**Quality:**

2

**Strengths And Weaknesses:**

Strengths:
* The method has good visual results compared to other methods, and can perform both inpainting and editing.
* The approach can also be extended to both flow-based and denoising-based approaches.

Weaknesses:
1. The paper has typographical and mathematical notation errors, both in main paper and in appendix. Please go over them. Some suggestions to improve:

* Section 3.1 is too long; please shorten it and use the freed space for an EM section.
* Notations are confusing across the paper. For instance, clarify that $ f_{\theta}(x_t) $ in Line 100 is Tweedie’s estimation, mapping $ x_t $ to $ x_0 $. In Equation 8, $ \mu $ looks like the denoising network output, so $ f_{\theta} $ and $ \mu $ should share a common symbol.
* In Figure 2, consider merging the two denoising-step blocks and explicitly show where latent masking occurs.
* Line 151 and Equation 10 state that EM comes from differentiating Equation 2 with respect to $ x_t $ and setting the result to zero. Please explain this derivation more clearly.
* Figure 3 is puzzling because the horse is masked out and replaced with unrelated content. What do we mask out and replace it with? Show the intended outputs of the VVAE decoding stage to clarify.

2. Even though the method looks promising at first, CD-EM and LMF is not justified adequately. CD-EM can be replaced with latent optimization, and LMF can be replaced with pixel-space masking, without a sub-network to train. Then, the overall method becomes a marginal improvement over DDNM. The authors should resolve the related issues in Questions part.

---

> ### Author Rebuttal · Authors · 2025-07-30
>
> > Weaknesses 1.a: Shorten the preliminary. Show where latent masking occurs in pipeline figure.
>
> Thanks for you advice. We are so pleased to update the paper to make it easier to read.
>
> > Weaknesses 1.b: Is there a confusion notation on $\boldsymbol{f}_\theta(\boldsymbol{x}_t)$ and $\boldsymbol{\mu} (\mathbf{x}_t, \boldsymbol{f} _\theta  (\boldsymbol{x}_t))$?
>
> In fact, there is no confusion. The notation of our paper follows the style of DDIM paper [3]. $\boldsymbol{f}_ \theta (\boldsymbol{x}_ t)$ estimates $\boldsymbol{x}_ 0 $. However, $\boldsymbol{u}(\boldsymbol{x}_ t , \boldsymbol{f}_ \theta(\boldsymbol{x}_ t))$ is the mean of $\boldsymbol{x}_ {t-1} $. As is introduced in line 95, it is a linear combination between $\boldsymbol{x}_ t$ and $\boldsymbol{f}_ \theta(\boldsymbol{x}_ t)$. The exact formulation is shown in Eqn.7 in DDIM paper. It is:
>
> $$\boldsymbol{\mu}(\boldsymbol{x}_ t, \boldsymbol{f}_ \theta(\boldsymbol{x}_ t)) = \sqrt{\alpha_ {t-1}} \boldsymbol{f}_ \theta(\boldsymbol{x}_ t) + \sqrt{1 - \alpha_ {t-1} - \sigma_t^2} \cdot \frac{\boldsymbol{x}_ t - \boldsymbol{f}_  \theta(\boldsymbol{x}_ t)}{\sqrt{1 - \alpha_ t}}$$
>
> We find readers may be confused with the $\boldsymbol{\mu}_ \theta(\boldsymbol{x}_ t)$ in Eqn.10. As is introduced in line 98, we define $\boldsymbol{\mu}_ \theta(\boldsymbol{x}_ t) = \boldsymbol{\mu}(\boldsymbol{x}_ t, \boldsymbol{f}_ \theta(\boldsymbol{x}_ t))$. We will update this part to make our notation more clear.
>
> > Weaknesses 1.c: Explain how the expectation in Eqn.8 is maximized. Is Eqn.10 differentiating Eqn.8 w.r.t. $\boldsymbol{x}_ t$?
>
> Eqn.8 defines a least square problem. Since $\|(\boldsymbol{x}_ {t-1} ^ a, \boldsymbol{x}_ {t-1}^ {b, i}) - \boldsymbol{\mu}(\boldsymbol{x}_ t, \boldsymbol{f}_ \theta(\boldsymbol{x}_ t))\|_ 2 ^ 2 \ge 0$, the term is minimized if and only if $(\boldsymbol{x}_ {t-1}^a, \boldsymbol{x}_ {t-1}^{b, i}) - \boldsymbol{\mu}(\boldsymbol{x}_ t, \boldsymbol{f}_ \theta(\boldsymbol{x}_ t))=0$. With this, we can already define a fixed point iteration problem. In our appendix, we use the analytical solution for least square problem. The solution is shown in Eqn.4 in the appendix. We do not compute the derivative w.r.t. $\boldsymbol{x}_ t$ in our paper.
>
> > W 1.d: Latent mask fuser illustration is using two irrelevant videos.
>
> This is very good advice! In practice, we use mask to fuse the model generated masked region with the known background. We will update it for better readability.
>
> > W 2.a: Why not use latent optimization?
>
> We tried to optimize the latent throught back-prop with the objective Eqn.9. Below, we show a quantitative comparison on DAVIS video inpainting. We use an adaptive learning rate $\eta_t = 0.02 \sqrt{\alpha_t}$ suggested by COPAINT. The experiment shows, fixed point iteration achieves better performance while using considerable lower cost.
>
> **Computation and memory cost**
> One objective of our method is to run it without using more graphic memory than normally sampling videos from the model. As the text-to-video generators becomes larger and video resolution becomes higher, this problem becomes more severe. Latent optimization method requires one or multiple back-propagation of the diffusion model (with gradient checkpointing, in most cases, which leads to even more cost), this introduces significantly more computation and memory usage. As shown below.
>
> |Method|NFEs|Back-props|PSNR $\uparrow$|SSIM $\uparrow$|VFID $\downarrow$|$E_{\mathrm{warp}} \downarrow$|Time|Memory|
> |-|-|-|-|-|-|-|-|-|
> |Latent optimization|50|25|34.03|0.9756|0.115|1.217|292.6 s|76,354 MB|
> |Latent optimization|50|50|34.11|0.9759|0.107|1.203|457.7 s|76,362 MB|
> |Latent optimization|50|75|34.13|0.9761|0.101|1.185|619.3 s|76,364 MB|
> |Fixed point iteration (ours)|100|0|34.13|0.9765|0.096|1.163|249,4 s|22,569 MB|
>
> > Weaknesses 2.b: Why not use pixel space masking? Comparison between LMF and direct masked replacing in latent space.
>
> This is a pity we cannot show you the visual comparison due to the full-text rebuttal this year. A quantitative comparison on DAVIS is shown below. Pixel space masking is prohibitively expensive for latent video diffusion models, as it requires decoding to pixels, applying the mask, and re-encoding to latent space at every denoising iteration. Video VAE encoding/decoding alone takes ~6 seconds per iteration, making this approach impractical.
>
> Our Latent Mask Fuser (LMF) eliminates this overhead, reducing latency to 0.07s versus 5.67s for pixel-space masking while maintaining near-identical quality (PSNR: 34.49 vs. 34.67; SSIM: 0.9774 vs. 0.9789). Direct latent-space mask replacement (0.01s) causes severe boundary artifacts due to spatial-temporal compression (e.g., 4×8×8 downsampling in CogVideoX), and the video latent is somewhat delicate. It degrade PSNR to 31.64 and SSIM to 0.9578.
>
> |Method|Time|PSNR $\uparrow$|SSIM $\uparrow$|
> |-|-|-|-|
> |LMF|0.07s|34.49|0.9774|
> |Latent space mask fusing|0.01s|31.64|0.9578|
> |Pixel space mask fusing|5.67s|34.67|0.9789|
> > Question 1: Compared to RePaint and COPAINT theoretically
>
> In the paper of RePaint. The key algorithm is described in Eqn.(8a), Eqn.(8b) and Eqn.(8c). This is identical to back projection method described in DDNM [1]. Its limitation is shown in line 126 in our paper.
>
> COPAINT proposes to optimize latent $\boldsymbol{x}_t$ by maximizing the conditional posterior in each denoising step. As Eqn.10 in COPAINT paper says, we translate the notation to the style of our paper:
>
> $$\boldsymbol{x}_ t ^ * = \arg \min_{\boldsymbol{x}_ t} \|\boldsymbol{x}_ 0 ^ a - \boldsymbol{f}_ \theta ^ a (\boldsymbol{x}_ t)\|_ 2 ^ 2$$ where $\boldsymbol{f}_ \theta ^ a (\boldsymbol{x}_ t)$ is the unmasked part of model prediction. A foundamental difference between COPAINT objective and the EM objective is that, our objective includes the term $\boldsymbol{x}_ t$ instead of only $\boldsymbol{f}_ \theta ^ a (\boldsymbol{x}_ t)$. Only then, we can write the solution in this formation $\boldsymbol{x}_ t ^ * = g(\boldsymbol{x}_ t ^ *)$, and fixed point iteration can be applied.
>
> Note that we also extend the objective for conditional flow matching generative models, as shown in our appendix. Crucially, as reported by COPAINT in Fig.3, the latent optimization method requires siginicantly more time to compute than DDNM (about 5 to 20 times more depending on hyparameter choice). Please see W.2a to check how latent optimization is working in the video model. Our fixed-point iteration strategy is essential for efficient sampling.
>
> > Question 2: Visual illustration on convergency of the fixed point.
>
> We show how CD-EM reaches a fixed point by evaluating how much $\boldsymbol{f}_ \theta (\boldsymbol{x}_t)$ is different from the previous iteration. The similarity is evaluated using PSNR on 10 sampling videos. In this experiment, we use $N=1, P=1$. Note that, each row show the convergency of a certain denosing timstep. As K (iteration steps) becomes larger, the fixed point iteration will tend to not to change the content, and reaches a fixed point.
>
> |Steps (K)|t|1|2|3|5|10|15|20|30|
> |-|-|-|-|-|-|-|-|-|-|
> |PSNR $\uparrow$|5|17.54|21.36|24.84|27.84|28.64|29.31|29.68|29.81|
> |PSNR $\uparrow$|15|22.61|24.46|26.44|28.03|29.34|30.27|30.94|31.26|
> |PSNR $\uparrow$|25|26.41|28.81|29.69|30.54|31.21|31.85|32.24|32.41|
>
> > Question 3: A pseudo-inverse example for Eqn.5.
>
> To apply Eqn.5. One must flatten the image to a 1d vector (e.g, flatten using raster order). Now, I show the case where A is a 2x2 mean pooling operator on a 4x4 image. We have:
>
> $$\mathbf{A} = \left[\begin{matrix}
> 0.25 & 0.25 & 0 & 0 & 0.25 & 0.25 & 0 & 0 & 0 & 0 & 0 & 0 & 0 & 0 & 0 & 0 \\\\
> 0    & 0    & 0.25 & 0.25 & 0 & 0 & 0.25 & 0.25 & 0 & 0 & 0 & 0 & 0 & 0 & 0 & 0 \\\\
> 0 & 0 & 0 & 0 & 0 & 0 & 0 & 0 & 0.25 & 0.25 & 0 & 0 & 0.25 & 0.25 & 0 & 0 \\\\
> 0 & 0 & 0 & 0 & 0 & 0 & 0 & 0 & 0    & 0    & 0.25 & 0.25 & 0 & 0 & 0.25 & 0.25
> \end{matrix}\right].$$ Then the pseudo-inverse of it is a upsampling operator
>
> $$\mathbf{A}^ \dagger = \left[\begin{matrix}
> 1 & 1 & 0 & 0 & 1 & 1 & 0 & 0 & 0 & 0 & 0 & 0 & 0 & 0 & 0 & 0  \\\\
> 0    & 0    & 1 & 1 & 0 & 0 & 1 & 1 & 0 & 0 & 0 & 0 & 0 & 0 & 0 & 0 \\\\
> 0 & 0 & 0 & 0 & 0 & 0 & 0 & 0 & 1 & 1 & 0 & 0 & 1 & 1 & 0 & 0 \\\\
> 0 & 0 & 0 & 0 & 0 & 0 & 0 & 0 & 0    & 0    & 1 & 1 & 0 & 0 & 1 & 1
> \end{matrix}\right]^ T.$$
>
> > Question 4: Visual comparison with BP. CD-EM's improvements over BP.
>
> **Visual comparison with BP**
> We will add a visual ablation comparison to our paper. Due to the full-text rebuttal this year. We recommend looking to Fig.4 for a qualitative comparison for now (you are right only using BP is equal to DDNM).
>
> **CD-EM is the major contributor to video quality**
> In Tab.3, the four metrics are evaluating the performance from very different perspectives. In PSNR, and SSIM, the majority improvements are brought by BP because they are evaluating whether the background region of the inpainting results looks similar to the ground truth. BP is directly replacing the known background to the generated samples. This makes it naturally the most significant contributor to PSNR and SSIM. In contrast, CD-EM is mainly working on the masked region, which is not considered by PSNR and SSIM. As shown in Tab.3, CD-EM contributed the most on VFID and $E_{\mathrm{warp}}$, the video quality metrics. Therefore, CD-EM is in fact the major contributor to video quality.
>
> > Question 6: Shadow in the car-roundabout example.
>
> As shown in the first row of Fig.4, the official mask is not including the car shadow. Then all examples in the comparison is not removing the shadow. We rework it by inflating the car mask by 64 pixels to get an extended mask covering the shadow. Then our method is able to render that region. We are sorry for not being able to show you the visual result due to the regulation this year.
>
> [1] Zero-Shot Image Restoration Using Denoising Diffusion Null-Space Model
> [2] Video diffusion models
> [3] Denoising Diffusion Implicit Models

---

### Official Review · Reviewer_vt1q · 2025-06-26

**Clarity:** 2
**Significance:** 3
**Originality:** 3
**Rating:** 4
**Confidence:** 5

**Summary:**

The paper presents a novel, model-agnostic, training-free video inpainting and editing method that achieves strong performance and is supported by solid theoretical analysis. However, its reliance on multiple auxiliary models and limited comparison with other training-free methods raise concerns about pipeline robustness and experimental completeness.

**Questions:**

Questions
1. Could the authors provide additional comparisons with recent training-free video editing methods, such as AnyV2V or ReVideo, to better contextualize the performance of ZeroPatcher?
2. Given the reliance on accurate captions and object descriptions (e.g., from LLaVA-NeXt or GPT-4 mini), how robust is the method when such textual inputs are noisy or imprecise? Have the authors evaluated the model's sensitivity to caption quality?
3. More questions please refer to the weakness.

**Ethical Concerns:**

["NO or VERY MINOR ethics concerns only"]

**Final Justification:**

The rebuttal has addressed the main issues I raised in my initial review. No further critical concerns remain. I therefore keep my evaluation unchanged at borderline accept.

**Limitations:**

Yes. The authors have clearly acknowledged the limitations of their approach and briefly discussed potential negative societal impacts. Their transparency is appreciated.

**Paper Formatting Concerns:**

No major formatting issues observed.

**Quality:**

3

**Strengths And Weaknesses:**

Strengths
1. The paper introduces ZeroPatcher, a novel training-free method for video inpainting and editing, which leverages pre-trained text-to-video foundation models to produce high-quality results without any additional training.
2. The proposed method is model-agnostic and can be flexibly applied to a range of diffusion-based video generators, including CogVideoX and HunyuanVideo.
3. Extensive experiments demonstrate the effectiveness of ZeroPatcher in both video inpainting and editing tasks, showing superior performance over existing training-free baselines and even outperforming some trained models on certain metrics.
4. The paper presents a thorough theoretical analysis of the proposed Conditional Denoising Expectation Maximization (CD-EM) algorithm, including proofs of solution uniqueness and convergence under well-defined conditions.

Weaknesses
1. Although the method is described as training-free, it still relies on several auxiliary base models. For instance, the use of CogVideoX involves external components such as LLaVA-NeXt and GPT-4 mini (o1) for caption generation, which introduces potential fragility into the overall pipeline. Moreover, the latent mask fuser appears to require additional training, which somewhat compromises the training-free claim.
2. The paper lacks comparisons with several recent training-free video editing or inpainting methods, such as AnyV2V and ReVideo, which are relevant for a comprehensive evaluation.
3. Certain formatting issues are present throughout the manuscript, including inconsistent line spacing and layout irregularities (e.g., Line 200 and the caption of Table 2), which may affect readability.

---

> ### Author Rebuttal · Authors · 2025-07-31
>
> > W 1: Is our method fragile for using the captioner? Is our method training-free with LMF?
>
> **Usage of captioner**
>
> We are giving text-to-video generators the ability for video inpainting and video editing. As the feature of a text-to-video generators, the model depends a lot on the input prompts. Luckily, the LLavaNext + GPT4-o captioning strategy works well for our experiments. We first use LLavaNext to caption the video and extract the main objective within the mask.
>
> For example, in the car-roundabout video, the original prompt is: "The video captures a sequence of images showing a dark-colored Mini Cooper car in motion on a city street. The vehicle is positioned at different angles, indicating it is turning or navigating through the traffic. The backdrop includes buildings with classical architecture, parked cars, and blue parking signs. The sky is clear, suggesting fair weather conditions."
>
> We use GPT4-o to image a video without the masked object to produce the inpainting prompt. "The video shows a city street with buildings, parked cars, and a blue traffic sign. The sky is clear, suggesting fair weather conditions." The performance for removing the main objective from video caption is satisfying for DAVIS and Youtube-VOS. We believe during the practical use of ZeroPatcher, a real user will not be troubled by this prompt dependency.
>
> **Is our method strictly training-free**
>
> The usage of LMF will not compromise our training-free claim. The main reasons are:
> 1. LMF is our computation optimization patch for latent video diffusion models. If ZeroPatcher is applied to pixel space video diffusion models. There is no need for LMF.
> 2. For latent video diffusion models. Our method can achieve the same performance at the cost of additional computation for mask fusing operation. LMF is the way to erase the additional cost.
>
> There are three ways to achieve mask replacing opeartion for latent video diffusion models:
> + Method 1: Directly downsample pixel space mask with the scale 4x8x8. Then, we apply the downscaled mask for direct latent space masked replacing.
> + Method 2: First, decode the latent to pixel videos using VAE decoder. Then, apply pixel space masked replacing. Finally, encode the replaced result to latent space using VAE encoder.
> + Method 3: Train and use LMF.
>
> To compare these three methods. We randomly sample 50 video pairs from DAVIS, each pair use a randomly sampled video mask. The methods are used to fuse the video pairs. The ground truth is obtained by using masked replacing in pixel space.
>
> |Method|Time|PSNR $\uparrow$|SSIM $\uparrow$|
> |-|-|-|-|
> |1. Latent space mask fusing|0.01s|31.64|0.9578|
> |2. Pixel space mask fusing with VAE|5.67s|34.67|0.9789|
> |3. LMF|0.07s|34.49|0.9774|
>
> As shown by the table. Method 2 achieves very good masked replacing performance. It requires no training, but will use a lot additional computation. One can replace LMF with method 2 to achieve strict training-free.
>
> > W 2: Comparison with training-free video editting method such as AnyV2V and ReVideo.
>
> AnyV2V is a training-free editing method that improves over the framework VideoP2P. ReVideo, however, is a requires the training of a ControlNet-like architecture to inject content and motion. We compare ZeroPatcher with them on DAVIS using automatic metrics for a comprehensive evaluation. The result is shown below:
>
> |Method|Trainiing-free|CLIP score $\uparrow$|PSNR $\uparrow$|SSIM $\uparrow$|$E_{warp}$ $\downarrow$|
> |-|-|-|-|-|-|
> |AnyV2V|Y|0.3396|31.07|0.9467|1.354|
> |ReVideo|N|0.3494|34.12|0.9703|1.259|
> |Ours|Y|0.3543|34.23|0.9712|1.231|
>
> > W 3: Formating problem.
>
> We update our script for improved readability. Thanks for you advise.
>
> > Q 2: Sensitivity to caption quality.
>
> It is very valuable to introduce a caption quality sensitive analysis in our paper. We are working on it.

---

### Official Review · Reviewer_PvBK · 2025-07-02

**Clarity:** 3
**Significance:** 3
**Originality:** 3
**Rating:** 5
**Confidence:** 4

**Summary:**

This paper presents a model-agnostic method for video inpainting and editing that can be integrated into existing generative video frameworks without requiring additional training. The authors propose a Latent Mask Fuser to enable accurate masking in the latent space, avoiding explicit VAE encoding/decoding. Additionally, they introduce Conditional Denoising Expectation Maximization (CD-EM), which formulates the generative process as an EM problem conditioned on the observed data. Supplementary materials demonstrate the method’s superiority over prior work across video inpainting and editing tasks.

**Questions:**

While the method’s increased inference cost is a concern, its architectural flexibility and empirical performance are compelling. Including additional qualitative ablation and more relevant baselines (e.g., VACE) would make the work even stronger.

**Ethical Concerns:**

["NO or VERY MINOR ethics concerns only"]

**Final Justification:**

- The proposed method presents a practical trade-off between computational cost and performance.

- The use of the EM-based approach offers a promising direction that could inspire other training-free methods, especially in the era of large models.

- Open-sourcing the WAN-based version would be valuable for the research community, particularly for advancing video editing and generation tasks.

**Limitations:**

yes

**Quality:**

3

**Strengths And Weaknesses:**

Pros:

- The proposed approach is designed to be compatible with a variety of diffusion and flow matching based text-to-video generation models, improving its versatility and practical applicability.
- The method shows clear improvements in both video inpainting and editing tasks, as demonstrated by qualitative and quantitative comparisons with prior methods.

Cons:
- Due to the EM formulation, the method introduces significant inference-time overhead, as reported in the appendix. Discussion on potential optimization or trade-offs would strengthen the paper.
- While the proposed components appear effective, visual ablation studies would be helpful in further clarifying their individual contributions.
- It would be valuable to evaluate the method on top of WAN 2.1, and compare against VACE [a], a state-of-the-art editing model built on WAN 2.1. Such a comparison would evaluate performance under similar architectures and clarify the benefits of the training-free method.

[a] VACE: All-in-One Video Creation and Editing

---

> ### Author Rebuttal · Authors · 2025-07-31
>
> > W 1: Inference speed overhead, possile trade-offs or optimization.
>
> **Inference speed overhead**
> Please refer to our response to W 1 for Reviewer U6KC. Our method is not considerably slower than the editing and inpainting baselines. For your convenience, we copy it here
>
> > Compared to the baseline DDNM, both DDNM (using 100 inference timesteps) and ZeroPatcher (50 timesteps, for the first 25 steps, use CD-EM with $N=1, K=1, P=3$) uses 100 NFEs. DDNM consumes 241.7 seconds, and ZeroPatcher 250.9 seconds for video inpainting. Our method is not using significantly more time to run compared to the training-free inpainting baseline.
> For video editing, VideoComposer and VideoP2P are running on smaller resolution (VideoComposer: 16x256x256, VideoP2P: 8x512x512) compared to ours (49x720x480). If we simply use them to perform 49-frame editing clip by clip, our method is in fact not using more time than them.
> Meanwhile, it would be unfair to compare ZeroPatcher (250.9 seconds) with the text-to-video foundation model (78.3 seconds) on inference speed, since the foundation model do not have video inpainting and editing capability.
>
> **Possible trade-offs or optimization**
>
> One can choose to use smaller $P, K, N$ to achieve faster sampling. An extreme situation is when $P = 1, K = 1, N = 1$, we reduce the NFEs to 75. The inference time is reduced to 191.4 seconds (22% speed up). As Fig.6 shows, RFID will degrade to 0.102.
>
> > W 2: Visual ablation studies.
>
> Due to the text-only rebuttal regulation this year, we are sorry the visual ablation cannot be shown to you now. We are additing it to the appendix. In alternative, we provide some evidences on how each tecnical component contributes to the final result.
>
> **Contribution of CD-EM**
> When the model uses PF only, it is identical to DDNM. In Fig.4, we show a visual comparison between DDNM and ZeroPatcher. It is showing what the results look like without CD-EM.
>
> **Contribution of LMF**
> When not using LMF, the model adopts direct latent space mask fusing. Given a dynamical video mask in the pixel space. We first downsample it using the scale 4x8x8. Then, we apply masked replacing in latent space directly. Below, we show the performance comparison on LMF and the naive method. We randomly sample 50 video pairs from DAVIS, each pair use a randomly sampled video mask. The methods are used to fuse the video pairs. The ground truth is obtained by using masked replacing in pixel space. It shows LMF achieves much better fusing performance.
>
> We cannot show visual comparison due to the full-text rebuttal rule this year. But if you look into the results, latent space mask fusing is doing okay in the centeral area of the mask. But very severe artifacts are observed in mask boundaries.
>
> |Method|PSNR $\uparrow$|SSIM $\uparrow$|
> |-|-|-|
> |LMF|34.49|0.9774|
> |Latent space mask fusing|31.64|0.9578|
>
> > W 3: ZeroPatcher on Wan 2.1.
>
> This is very good advice to adopt ZeroPatcher on Wan 2.1. We are now working on it and ZeroPatcher + Wan 2.1 will be available when we open-source our project. Then, we can add a comparison between ZeroPatcher and VACE.

---

> ### Comment · Reviewer_PvBK · 2025-08-04
> **Thanks for the rebuttal**
>
> Thank you to the authors for preparing the rebuttal. I have also read the other reviews. My concerns have been well addressed, and I find the additional comparisons with latent optimization methods to be particularly impressive. The proposed approach offers a practical trade-off between computational cost and performance. I would like to raise my rating.
>
> The WAN-based variant and the comparisons with VACE are valuable contributions that could benefit the research community. Additionally, as noted by other reviewers, clarifying the relationship between the latent mask fuser training and the training-free claim is important for understanding the method's scope and limitations.

---

### Official Review · Reviewer_U6KC · 2025-07-03

**Clarity:** 3
**Significance:** 3
**Originality:** 4
**Rating:** 5
**Confidence:** 3

**Summary:**

The authors introduce ZeroPatcher, a training-free sampling algorithm that adapts video diffusion models for video inpainting and editing. Specifically, the authors introduce CD-EM to optimise the masked area to match the likelihood of the unmasked regions through a fixed-point EM strategy. The authors also introduce a latent mask fuser that allows them to directly mask at the latent space instead of the pixel space. The authors evaluate their training-free strategy across various models and datasets/tasks and find that their model performs on par with the state-of-the-art training-based model.

**Questions:**

In the order of importance (not same as the order of ease):

1. Would you be releasing the code?
2. How does the model perform under different masking conditions, specifically extreme masking conditions?
3. How does the inference speed change with the different values of $P, N, K$?
4. Can you provide additional details about the human evaluation performed?

**Ethical Concerns:**

["NO or VERY MINOR ethics concerns only"]

**Final Justification:**

The rebuttal has sufficiently addressed my queries with the paper. With that done, the paper is suitable for acceptance. Therfore I will retain my score as 5.

**Limitations:**

1. Improved editing ability can lead to a potential negative societal impact. Follow-up works could explore solving EM with additional steering/alignment to prevent generating unwanted content.

**Paper Formatting Concerns:**

Reference 32 is exceeding the page margins in the main text. The same reference is exceeding the page margins in the appendix.

**Quality:**

4

**Strengths And Weaknesses:**

### Strengths:

1. The paper is well written, and the content is easy to follow.
2. The method appears to be theoretically sound and achieves good results.
3. Since it’s a training-free algorithm, it would allow for easy adoption and development due to its reduced requirements.

### Weaknesses:

1. The added steps of Expectation-Maximisation make the approach slower compared to the baselines.
2. Appendix D: “ZeroPathcer” vs “ZeroPatcher”
3. The notation in eq 8, eq 9, eq 10, eq 11 with $(x_1, x_2)$ seems to be unconventional and confusing.

---

> ### Author Rebuttal · Authors · 2025-07-31
>
> Thanks for your professional review. The considerations are very valuable in making our paper better. The typos are now corrected in our script. Our response is shown below.
>
> > W 1: Inference speed compared with baselines.
>
> Compared to the baseline DDNM, both DDNM (using 100 inference timesteps) and ZeroPatcher (50 timesteps, for the first 25 steps, use CD-EM with $N=1, K=1, P=3$) uses 100 NFEs. DDNM consumes 241.7 seconds, and ZeroPatcher 250.9 seconds for video inpainting. Our method is not using significantly more time to run compared to the training-free inpainting baseline.
>
> For video editing, VideoComposer and VideoP2P are running on smaller resolution (VideoComposer: 16x256x256, VideoP2P: 8x512x512) compared to ours (49x720x480). If we simply use them to perform 49-frame editing clip by clip, our method is in fact not using more time than them.
>
> Meanwhile, it would be unfair to compare ZeroPatcher (250.9 seconds) with the text-to-video foundation model (78.3 seconds) on inference speed, since the foundation model do not have video inpainting and editing capability.
>
> > W 3: notation in Eqn.8, Eqn.9, Eqn.10, and Eqn.11
>
> Thanks for the advice. We are pleased to improve our notation to make it easier to follow. We replace the notation $(\boldsymbol{x}_ {t - 1} ^ a, \boldsymbol{x}_ {t-1} ^ {b, i})$ with $\tilde{\boldsymbol{x}}_ {t-1} = \\{\boldsymbol{x}_ {t - 1} ^ a, \boldsymbol{x}_  {t-1} ^ {b, i}\\}$. We will explain the bracket means masked fusing as shown in Eqn.16. It is also consistent to our notation in Eqn.16.
>
> > Q 1: Release code.
>
> Yes, we are now working on it. The project will open-source soon.
>
> > Q 2: Performance on extreme masking conditions.
>
> It is a valueable to test our method on extreme masking conditions. We are working on the outpainting and large region inpainting test. The visual results will be added to the appendix.
>
> > Q 3: How inference speed change according to P, K, N
>
> In Tab.1 in the appendix. We show a theoretical boundary of computation complexity given $P, K, N$. We use $T$ timesteps to sample a video among them, the first $\tau$ steps will use CD-EM. The NFEs of ZeroPatcher can be computed using the following formula.
> $$\mathrm{NFEs} = \tau (PN + P(K-1)) + (T - \tau), \qquad P \ge 1, N \ge 1, K \ge 1, \tau \le T$$ Below, we show the measured inference speed ($T = 50, \tau = 25$).
>
> |P|K|N|NFEs|Time|
> |-|-|-|-|-|
> |3|1|1|100|259.9 seconds|
> |3|1|2|175|454.8 seconds|
> |3|2|1|175|457.1 seconds|
> |5|1|1|150|388.6 seconds|
> |5|1|2|275|718.3 seconds|
> |5|2|1|275|720.1 seconds|
>
> > Q 4: Human evaluation details.
>
> We create a questionaire containing 50 questions using all video editing samples computed by all methods on DAVIS (But skip VideoP2P on background consistency, because it do not preserve background, comparing it is meaningless). Each question shows the editing results from all 5 methods along with the editing input. The question ask the respondent to answer which one achieves the best background preservation, and which one matches the editing prompt most. The editing results are labeled anonymously (By A, B, C, D, E).
> During the human evaluation, the participant will randomly receive 1 of the 50. The labeling and display order of the editing results are shuffled each time we show the participant the question. We ensure the participant will not receive repeated questions. the participant can answer up to 50 questions. We collect 1,310 answers from 171 different participants.

---

> > ### Comment · Reviewer_U6KC · 2025-08-04
> > **Thanks for the rebuttal**
> >
> > Dear Authors,
> >
> > After reading the rebuttal and the feedback from the other reviewers, I believe that my queries have been answered sufficiently. The authors should update the manuscript with the improved notations and info about W1, Q2, Q3 and Q4. Additional experiments about the latent mask fuser also seem interesting -- I request that the authors clarify that as well in the final manuscript, since that might be an outstanding contribution to the community. I also hope that the authors provide the code as promised.
> >
> > Best,

---

### Decision · Program_Chairs · 2025-09-17

**Decision:**

Accept (poster)

**Comment:**

The authors introduce ZeroPatcher, a novel training-free model-agnostic sampler for video inpainting and editing using pre-trained text-to-video foundation models.

The sampler optimizes the missing/masked parts of the denoising trajectory by maximising the (marginal) log-likelihood expectation conditioned on unmasked video segments. The inner-loop marginalization is handled by applying the Expectation-Maximization (EM) problem over the denoising trajectory, involving an expectation step via Monte-Carlo sampling and a maximization step solved by fixed-point iteration, with theoretical analysis showing solution uniqueness and convergence. CD-EM conditions on the unmasked part of x_{t-1} as guidance, which is more informative than using the unmasked x_t as is done by traditional back-projection methods.
To enable inpanting and editing with latent-space video models without repeatedly converting between the latent and pixel spaces, the authors introduce a lightweight convolutional architecture called the Latent Mask Fuser. This module performs video masking directly in latent space, thereby eliminating the need for VAE decoding and encoding, which are computationally expensive.
ZeroPatcher demonstrates impressive video inpainting performance using CogVideoX and HunyuanVideo as foundation models. It outperforms the training-free inpaining baselines and is even competitive with trained inpainting models.

Strengths:
- The proposed approach is novel, sensible, well-motivated, and clearly described.
- It performs well at both video inpainting and editing, and its training-free and model-agnostic nature makes it widely applicable and well positioned to benefit from improvements in the underlying foundation models.
- Additional results provided by the authors in response to the reviewers strengthen the paper further.

Weaknesses:
- While some reviewers pointed out that the fact that the Latent Mask Fuser (LFM) needed to be trained contradicted the training-free label of the method, the authors clarified that the LFM was used to improve efficiency of the approach by eliminating the repeated conversion between the latent and pixels spaces, and that the sampler itself is truly training-free and can be used without LFM and provided empirical evidence of this claim.
- As ZeroPatcher uses EM in its sampling loop, it is going to require more computation than vanilla sampling from the underlying foundation model. However, hyperparameters (N,P, and K) provide a way of trading off the quality of the results against the additional computation and the authors show that good results can be obtained at a relatively low additional computations cost.
- One of the reviewers had concerns about the notation, but I found nothing nothing wrong with it.
- Some of the more mathematical parts of the paper (e.g. Theorem 3.1) are harder to follow than necessary because of choppy writing. Please merge consecutive short sentences into longer sentences when they clearly belong together.

This paper makes a substantial contribution to video generation and editing, and clearly meets the bar for acceptance.